# DORSal: Diffusion for Object-centric Representations of Scenes *et al.*

**Allan Jabri**[†,*]
UC Berkeley

**Sjoerd van Steenkiste**[*]
Google Research

**Emiel Hoogeboom**
Google DeepMind

**Mehdi S. M. Sajjadi**
Google DeepMind

**Thomas Kipf**
Google DeepMind

## Abstract

Recent progress in 3D scene understanding enables scalable learning of representations across large datasets of diverse scenes. As a consequence, generalization to unseen scenes and objects, rendering novel views from just a single or a handful of input images, and controllable scene generation that supports editing, is now possible. However, training jointly on a large number of scenes typically compromises rendering quality when compared to single-scene optimized models such as NeRFs. In this paper, we leverage recent progress in diffusion models to equip 3D scene representation learning models with the ability to render high-fidelity novel views, while retaining benefits such as object-level scene editing to a large degree. In particular, we propose DORSal, which adapts a video diffusion architecture for 3D scene generation conditioned on frozen object-centric slot-based representations of scenes. On both complex synthetic multi-object scenes and on the real-world large-scale Street View dataset, we show that DORSal enables scalable neural rendering of 3D scenes with object-level editing and improves upon existing approaches.

## 1 Introduction

Recent works on 3D scene understanding have shown how geometry-free neural networks trained on a large number of scenes can learn scene representations from which novel-views can be synthesized (Sitzmann et al., 2021; Sajjadi et al., 2022c). Unlike Neural Radiance Fields (NeRFs) (Mildenhall et al., 2020), they are trained to generalize to novel scenes and require only few observations per scene. They also benefit from the ability of learning more *structured* scene representations, e.g. object representations that capture shared statistical structure (e.g. cars) observed throughout many different scenes (Stelzner et al., 2021; Yu et al., 2022; Sajjadi et al., 2022a). However, these models are trained with only a few observations per scene, and without a means to account for the uncertainty about scene content that remains unobserved they typically fall short at synthesizing precise novel views and produce blurry renderings (see Figure 4 for representative examples).

Equally recently, diffusion models (Sohl-Dickstein et al., 2015) have led to breakthrough performance in image synthesis, including super resolution (Saharia et al., 2022c), image-to-image translation (Saharia et al., 2022a) and in particular text-to-image generation (Saharia et al., 2022b). Part of the appeal of diffusion models lies in their simplicity, scalability, and steer-ability via conditioning. For example, text-to-image models can be used to edit scenes via prompting because of the compositional scene structure induced by training with language (Hertz et al., 2023). While diffusion models have recently been applied to novel-view synthesis, scaling to complex visual scenes while maintaining 3d consistency remains a challenge (Watson et al., 2023).

---

[†]Work done while interning at Google, [*]equal contribution.
Correspondence: svansteenkiste@google.com, tkipf@google.com

In this work, we combine techniques from both of these subfields to further neural 3D scene rendering. We leverage frozen object-centric scene representations to condition probabilistic diffusion decoders capable of synthesizing novel views while also handling uncertainty about the scene. In particular, we use Object Scene Representation Transformer (OSRT) (Sajjadi et al., 2022a) to compute a set of *Object Slots* for a visual scene from only few observations, and condition a video diffusion architecture (Ho et al., 2022c) with these slots to generate sets of 3D consistent novel views of the same scene. We show that conditioning on *object-level* representations allows for scaling more gracefully to complex scenes, large sets of target views, and enables basic object-level scene editing by removing slots or by transferring them between scenes.

In summary, our contributions are as follows:

- We introduce *Diffusion for Object-centric Representations of Scenes et al.* (DORSal), an approach to controllable 3D novel-view synthesis combining (frozen) object-centric scene representations with diffusion decoders.

- Compared to prior methods from the 3D scene understanding literature (Sajjadi et al., 2022a;c), DORSal renders novel views that are significantly more precise (e.g. 5x-10x improvement in FID) while staying true to the content of the scene. Compared to prior work on 3D Diffusion Models (Watson et al., 2023), DORSal scales to more complex scenes, performing significantly better on real-world Street View data.

- Finally, we demonstrate how, by conditioning on a structured, object-based scene representation, DORSal learns to compose scenes out of individual objects, enabling basic object-level scene editing capabilities at inference time.

## 2 PRELIMINARIES

DORSal is a diffusion generative model conditioned on a simple object-centric scene representation.

**Object-centric Scene Representations.** Core to our approach to scene generation is the use of (pre-trained) object representations as conditioning information, as opposed to, e.g., conditioning on language prompts (Ramesh et al., 2021; Rombach et al., 2022; Ho et al., 2022c). Recent breakthroughs in neural rendering have inspired multiple works for learning such 3D-centric object representations, including uORF (Yu et al., 2022) and ObSuRF (Stelzner et al., 2021). However, these methods do not scale beyond simple datasets due to the high memory and compute requirements of volumetric rendering. More recently, the Object Scene Representation Transformer (OSRT) (Sajjadi et al., 2022a) has been proposed as a powerful method that scales to much more complex datasets with wider camera pose distributions such as MultiShapeNet (Sajjadi et al., 2022c). Building upon SRT (Sajjadi et al., 2022c), it uses light-field rendering to obtain speed-ups by a factor of $\mathcal{O}(100)$ at inference time. We use OSRT as a base model for obtaining object representations as conditioning information for DORSal.

An overview of OSRT's model architecture is shown in Figure 1(a). A small set of *input views* is encoded through a CNN followed by a self-attention Transformer (Vaswani et al., 2017) (*Encoder*). The resulting set-latent scene representation (SLSR) is fed to Slot Attention (Locatello et al., 2020), which cross-attends from a set of slots into the SLSR. This leads to the Object Slots, an object-centric description of the scene. The number of slots is chosen by the user and sets an upper bound on the number of objects that can be modeled for each individual scene during training.

Once the input views are encoded into the Object Slots, arbitrary novel views can be rendered by passing the target ray origin and direction (the *Pose*) into the *Decoder*. To encourage an object-centric decomposition in the Object Slots, Spatial Broadcast Decoders (Watters et al., 2019) are commonly used in the literature: Each slot is decoded independently into a pair of RGB and alpha using the same decoder, after which a Softmax over the slots decides on the final output color. Since OSRT is trained end-to-end with the L2 loss, any uncertainty about novel views necessarily leads to blur in the final renders. OSRT can be trained fully unsupervised (in the absence of object labels) or using segmentation supervision (Prabhudesai et al., 2023) to guide the decomposition process.

**Generative Modeling with Conditional DDPMs.** Denoising Diffusion Probabilistic Models (DDPMs) learn to generate data $x$ by learning the reverse of a simple destruction process (Sohl-

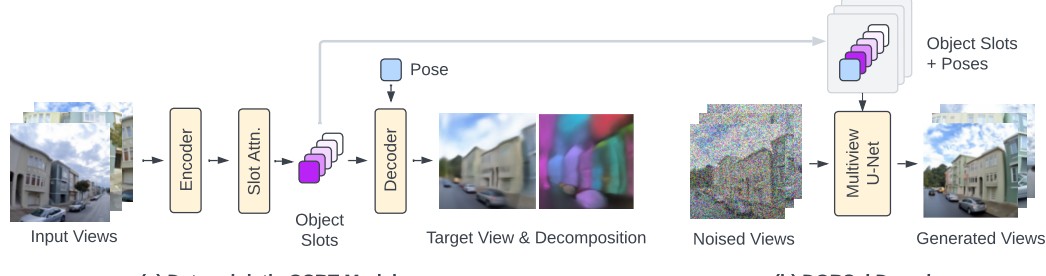

Figure 1: **Model overview.** (a) OSRT is trained to predict novel views through an Encoder-Decoder architecture with an *Object Slot* latent representation of the scene. Since the model is trained with the L2 loss and the task contains significant amounts of ambiguity, the predictions are commonly blurry. (b) After training the OSRT model, and freezing it, we take the Object Slots and combine it with the target Poses to be used as conditioning. Our Multiview U-Net is trained in a diffusion process to denoise novel views while cross-attending into the conditioning features (see Figure 2 for details). This results in sharp renders at test time, which can still be decomposed into the objects in the scene to support edits.

Dickstein et al., 2015). Such a diffusion process is convenient to express in its marginal form:

$$q(\boldsymbol{z}_t|\boldsymbol{x}) = \mathcal{N}(\boldsymbol{z}_t|\alpha_t\boldsymbol{x}, \sigma_t^2\mathbf{I}), \tag{1}$$

where $\alpha_t$ is a decreasing function and $\sigma_t$ is an increasing function over diffusion time $t \in [0, 1]$. A neural network is then used to approximate $\boldsymbol{\epsilon}_t$, the reparametrization noise, to sample $\boldsymbol{z}_t$:

$$L = \mathbb{E}_{t\sim\mathcal{U}(0,1),\boldsymbol{\epsilon}_t\sim\mathcal{N}(0,\mathbf{I})}\Big[w(t)||\boldsymbol{\epsilon}_t - f(\boldsymbol{z}_t, t)||^2\Big], \tag{2}$$

where $f$ is a neural network and $\boldsymbol{z}_t = \alpha_t\boldsymbol{x} + \sigma_t\boldsymbol{\epsilon}_t$. There exists a particular weighting $w(t)$ for this objective to be a variational negative lower bound on $\log p(\boldsymbol{x})$, although in practice the constant weighting $w(t) = 1$ has been found to be superior for sample quality (Ho et al., 2020; Kingma et al., 2021). Because diffusion models learn to correlate the pixels in their generations, they are able to generate images with crisp details even if the exact location of such details is not entirely known. We follow the framework of *conditional* diffusion models, where conditioning information $\boldsymbol{s}$, such as text or, in our case, information about scene content, is provided to the neural network function $f(\boldsymbol{z}_t, t, \boldsymbol{s})$, e.g. implemented using a cross-attention in a U-Net (Ronneberger et al., 2015).

## 3 DORSAL

DORSal consist of two main components, illustrated in Figure 1. First, we encode a few context views into Object Slots using the encoder of a pre-trained Object Scene Representation Transformer (OSRT) (Sajjadi et al., 2022a). Second, we train a video diffusion architecture (Ho et al., 2022c) conditioned on these Object Slots to synthesize a set of 3D consistent renderings of novel views of that same scene.

### 3.1 DECODER ARCHITECTURE & CONDITIONING

**Architecture details.** The DORSal decoder uses a convolutional U-Net architecture as is conventional in the diffusion literature (Ho et al., 2020). To attain consistency between $L$ views generated in parallel, following Video Diffusion (Ho et al., 2022c), each frame has feature maps which are enriched with 2d convolutions to process information within each frame and axial (self-)attention to propagate information between frames (see also Appendix B.2). We refer to this as a Multiview U-Net in our setting as each frame corresponds to a separate view of a scene. DORSal relies on Object Slots for context about the scene, which avoids the cost of attending directly to large sets of conditioning features that are often redundant.

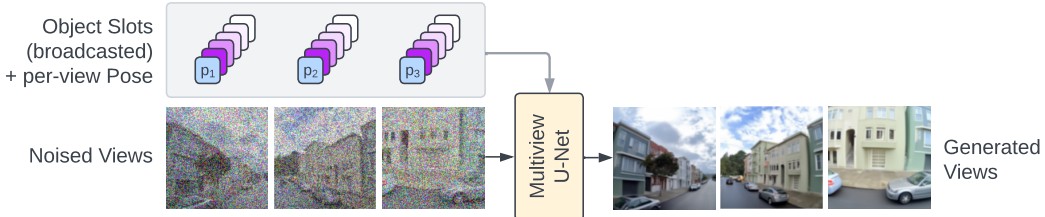

Figure 2: **DORSal slot and pose conditioning.** DORSal is conditioned via cross-attention and FiLM-modulation (Perez et al., 2018) on a set of Object Slots (shared across views) and a per-view Pose vector.

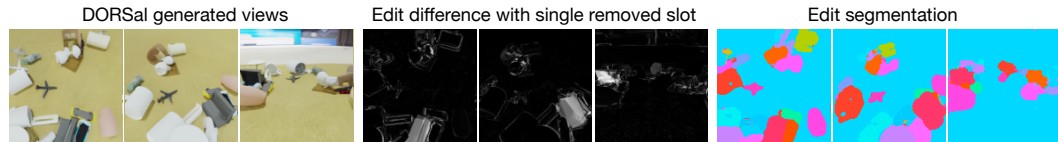

Figure 3: **DORSal scene editing and evaluation.** To obtain instance segmentations of objects in a scene, we perform scene edits by dropping out individual slots, rendering the resulting views, and computing a pixel-wise difference (*middle*) compared to the unedited rendered views (*left*). These differences are smoothed and thresholded to arrive at a segmentation image (*right*).

**Conditioning.**   The generator is conditioned with embeddings of the slots, target pose, and diffusion noise level. To compute these embeddings, given a set of $K$ Object Slots $[\mathbf{s}_1, \ldots, \mathbf{s}_K]$ that describe a single scene, we project the individual Object Slots and broadcast them across views. We append the target camera pose $\mathbf{p}_i$ to the Object Slots for each view $i = 1, \ldots, L$, after applying a learnable linear projection. Thus, each view $i$ is conditioned on the following set of $K + 1$ tokens: $[f(\mathbf{s}_1), \ldots, f(\mathbf{s}_K), g(\mathbf{p}_i)]$, where $f(...)$ and $g(...)$ are learnable linear projections to the same dimensionality $D$. This process is depicted in Figure 2.

We apply this conditioning in the same way that text is treated in recent work on text-to-image models (Saharia et al., 2022c), i.e. integrated into the U-Net in two ways: 1) we attention-pool (Radford et al., 2021) conditioning embeddings into a single embedding for modulating U-Net feature maps via FiLM (Perez et al., 2018), and 2) we use cross-attention (Vaswani et al., 2017) to attend on conditioning embeddings (keys) from the feature map (queries).

### 3.2 EDITING & SAMPLING

**Scene editing.**   At inference time, we explore a simple form of scene editing: by removing individual slots, we can—if the slot succinctly describes an individual object in the scene—remove that object from the scene. We remove slots by masking out the value of the slot, including any attention weights derived from it. Sampling with this edited conditioning yields $K$ edited scene renderings, where $K$ is the number of object slots in each model. We can then derive the effect of each edit by comparing it to unedited samples generated by keeping all slots for conditioning. To measure success, and to compare between methods, we propose to segment pixels based on whether they were affected by removing a particular slot, and compare to ground-truth instance segments using standard segmentation metrics. We further demonstrate successful transfer of objects between scenes as another form of scene editing.

To obtain instance segments from edits with DORSal, we propose the following procedure:

1. **Edit pixel difference:** We take the pixel-wise difference between unedited novel views and their edited counter-parts, averaged across color channels (see Figure 3 middle). This difference is sensitive to object removal if the revealed pixels differ in appearance from the removed object.

2. **Smoothing:** We apply a per-pixel softmax across all $K$ difference images to suppress the contribution of minor side effects of edits (e.g. pixels unrelated to an edited object that slightly change

after an edit) and provide a consistent normalization across each of the $K$ edits. Furthermore, we apply a median filter with a filter size of approx. 5% of the image size (e.g. width).

3. **Assignment:** Finally, we take the per-pixel argmax across $K$ edits to arrive at instance segmentation masks, from which we can compute segmentation metrics for evaluation.

**View-consistent camera path sampling.** Repeatedly generating blocks of $L$ frames is fast, but there is no guarantee on the consistency between the different blocks. This is because sampling from a conditional generative model inherently adds bits of information to the conditioning signal to produce a one-to-many mapping. Hence, achieving consistency across views involves synchronizing the manner in which bits of information are added, which is challenging as the number of output views grows beyond the amount used during training (as is required for generating long videos or smooth camera paths). We leverage the iterative nature of the generative denoising process to create *smooth transitions* as well as *global consistency* between frames. Our technique is inspired by Hoogeboom et al. (2023), where high resolution images are generated with overlapping patches by dividing the typical denoising process into multiple stages. For 3D camera-path rendering of hundreds of frames, we propose to interleave 3 types of frame shuffling for subsequent stages, while denoising only for a small number of steps per stage: 1) no shuffle (identity), to allow the model to make blocks of the context length consistent; 2) shift the frames in time by about half of the context length, which puts frames together with new neighbours in their context, allowing the model to create smooth transitions; 3) shuffle all frames with a random permutation, to allow the model to resolve inconsistencies globally.

## 4 RELATED WORK

**Novel View Synthesis (NVS) and 3D Scene Representations.** Motivated by NeRF (Mildenhall et al., 2020), significant advances have recently been achieved in neural rendering (Tewari et al., 2022). From many observations, NeRF optimizes an MLP through volumetric rendering, thereby allowing high-quality NVS. While several works extend this method to generalizing from few observations per scene (Yu et al., 2021; Chen & Xu, 2021), they do not provide accessible latent representations. Several *latent* 3D representation methods exist (Sitzmann et al., 2019; Eslami et al., 2018; Moreno et al., 2023), however they do not scale beyond simple synthetic datasets. The recently proposed Scene Representation Transformer (SRT, Sajjadi et al. (2022c)) and extensions (RUST, Sajjadi et al. (2022b)) use large set-latent scene representations to scale to complex real-world datasets with or without pose information. However, SRT often produces blurry images due the L2-loss and high uncertainty in unobserved regions. While approaches like Rombach et al. (2021) consider generative models for NVS, attaining 3d consistency is challenging with auto-regressive models.

**Diffusion Generative Models.** Modern score-based diffusion models (Sohl-Dickstein et al., 2015; Song & Ermon, 2019; Ho et al., 2020) have been very successful in multiple domains. They learn to approximate a small step of a denoising process, the reverse of the pre-defined diffusion process. This setup has proven to be very successful and easy to use compared to other generative approaches such as variational autoencoders (Kingma et al., 2021), normalizing flows (Rezende & Mohamed, 2015) and adversarial networks (Goodfellow et al., 2014). Examples where diffusion models have had success are generation of images (Ho et al., 2022b; Dhariwal & Nichol, 2021), audio (Kong et al., 2021), and video (Ho et al., 2022a). Moreover, the extent to which they can be steered to be consistent with conditioning signals (Ho & Salimans, 2021; Nichol et al., 2022) has allowed for much more controllable image generation. More recently, pose-conditional image-to-image diffusion models have been applied to 3D NVS (Watson et al., 2023; Liu et al., 2023; Gu et al., 2023; Chan et al., 2023; Tewari et al., 2023), focusing mainly on 3D synthesis of individual objects as opposed to complex visual scenes. Chan et al. (2023) presents results for some indoor scenes, though it remains unclear how to manipulate the generated scenes at the object level. DORSal leverages video diffusion models (Ho et al., 2022c) and object-slot conditioning to synthesize novel views that are more consistent, especially in real-world settings, and support object-level edits.

Object-centric methods have also been explored in combination with diffusion-based decoders: LSD (Jiang et al., 2023) and SlotDiffusion (Wu et al., 2023) combine Slot Attention with a diffusion decoder in latent space for image and (for the latter) video object segmentation. Neither approach, however, considers 3D scenes or NVS, but solely focus on auto-encoding objectives. In concurrent

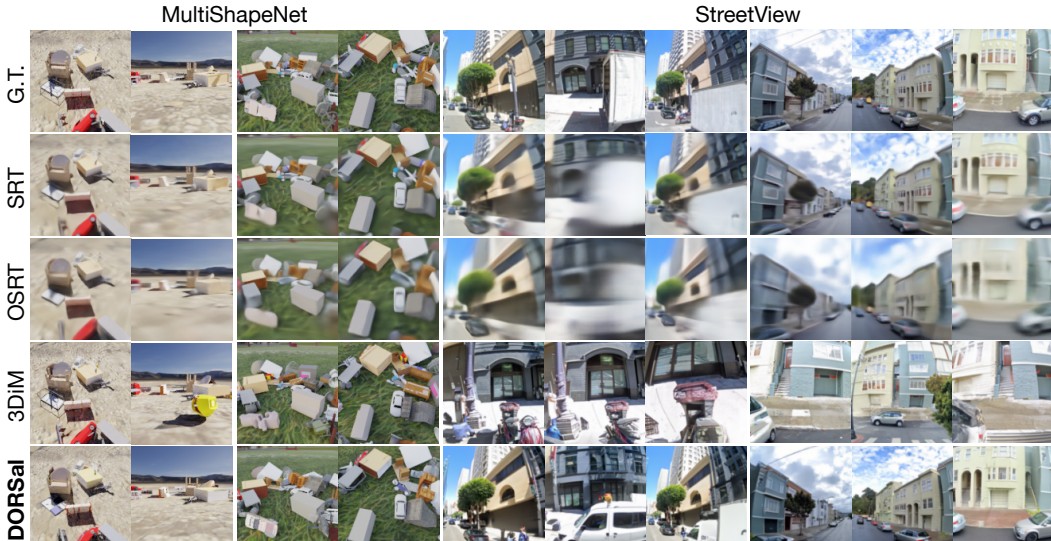

Figure 4: **Novel View Synthesis**. Comparison of DORSal with the following baselines: 3DiM (Watson et al., 2023), SRT (Sajjadi et al., 2022c), and OSRT (Sajjadi et al., 2022a) on the MultiShapeNet (only 2/5 views shown) and Street View datasets.

work, OBJECT 3DIT (Michel et al., 2023) finetunes a 3D diffusion model (Liu et al., 2023) on supervised object edits obtained from synthetically generated data of scene edits. In contrast to this work, scene edits afforded by DORSal do not require supervision or specifically prepared data of scene edits. Alternatively, DisCoScene (Xu et al., 2023), conditions adversarially-trained generators on an object-based scene layout to obtain a spatially disentangled generative radiance field from which a novel view can be rendered. In contrast, here we consider learned object representations as a conditioning signal, which further reduces the amount of prior knowledge needed about a scene.

## 5 EXPERIMENTS

We evaluate DORSal on challenging synthetic and real-world scenes in three settings: 1) we compare the ability to synthesize novel views of a scene with related approaches, 2) we analyze the capability for simple scene edits: object removal and object transfer between scenes, and 3) we investigate the ability of DORSal to render smooth, view-consistent camera paths. We provide detailed ablations in Appendix C.1. Complete experimental details are available in Appendix B and additional results in Appendix C.

**Datasets.** *MultiShapeNet (MSN)* (Sajjadi et al., 2022c) consists of scenes with 16–32 ShapeNet (Chang et al., 2015) objects each. The complex object arrangement, realistic rendering (Greff et al., 2022), HDR backgrounds, random camera poses, and the use of fully novel objects in the test set make this dataset highly challenging. We use the version from Sajjadi et al. (2022a) (*MSN-Hard*). The *Street View (SV)* dataset contains photographs of real-world city scenes. The highly inconsistent camera pose distribution, moving objects, and changes in exposure and white balance make this dataset a good test bed for generative modeling. Street View imagery and permission for publication have been obtained from the authors (Google, 2007).

**Baselines.** For comparison, we focus on SRT and OSRT from the 3D scene understanding literature (Sajjadi et al., 2022a;c), and 3DiM from the diffusion literature (Watson et al., 2023). Because OSRT (Figure 1(a)) and DORSal (Figure 1(b)) leverage the same object-centric scene representation, we can compare them in terms of the quality of generated novel-views as well as the ability to perform object-level scene edits. SRT, which was previously applied to Street View and mainly differs to OSRT in terms of its architecture, does not include Object Slots as a bottleneck. We use

Table 1: **Novel-view synthesis**. Comparing DORSal to methods based on scene representations.

| | MultiShapeNet | | | Street View | | |
|---|---|---|---|---|---|---|
| **Model** | **PSNR↑** | **LPIPS↓** | **FID↓** | **PSNR↑** | **LPIPS↓** | **FID↓** |
| SRT | **25.93** | **0.237** | 67.29 | **23.60** | **0.282** | 87.91 |
| OSRT | 23.35 | 0.330 | 100.7 | 21.19 | 0.410 | 165.1 |
| Sup-OSRT | 22.64 | 0.358 | 112.1 | — | — | — |
| DORSal | 18.76 | 0.266 | **11.01** | 16.05 | 0.361 | **16.24** |

Table 2: **Novel-view synthesis**. Comparing DORSal to 3DiM, here both methods use DDPM.

| | MultiShapeNet | | | Street View | | |
|---|---|---|---|---|---|---|
| **Model** | **PSNR↑** | **LPIPS↓** | **FID↓** | **PSNR↑** | **LPIPS↓** | **FID↓** |
| 3DiM | 18.20 | 0.287 | 10.94 | 12.68 | 0.477 | 15.58 |
| DORSal (DDPM) | **18.99** | **0.265** | **9.00** | **16.36** | **0.356** | **14.62** |

Sup-OSRT (Prabhudesai et al., 2023) to compute object-slots for DORSal on MultiShapeNet and plain OSRT on Street View (where ground-truth masks are unavailable).

3DiM is a pose-conditional image-to-image diffusion model for generating novel views of the same scene (Watson et al., 2023). During training, 3DiM takes as input a pair of views of a static scene where one of the views is corrupted with noise for training purposes. During inference, 3DiM makes use of *stochastic conditioning* to generate 3D-consistent views of a scene: a new view for a given target camera pose is generated by conditioning on a randomly selected view from a conditioning set at each denoising step. Each time a new view is generated, it is added to the conditioning set.

## 5.1 NOVEL-VIEW SYNTHESIS

**Set-up.** We separately train DORSal, OSRT, SRT, and 3DiM on MultiShapeNet and Street View, where DORSal and (Sup-)OSRT leverage the same set of Object Slots. We quantitatively evaluate performance at novel-view synthesis on a test set of 1000 scenes. We measure PSNR, which captures how well each novel view matches the corresponding ground truth, though is easily exploited by blurry predictions (Sajjadi et al., 2017). To address this we also measure FID (Heusel et al., 2017), which compares generated novel views to ground-truth at a distributional level, and LPIPS (VGG) (Zhang et al., 2018), which measures frame-wise similarities using deep feature embeddings.

**Results.** Quantitative results can be seen in Tables 1 & 2 and qualitative results in Figure 4. On MultiShapeNet and Street View it can be seen how DORSal obtains slightly lower PSNR compared to SRT and (Sup-)OSRT, but greatly outperforms these methods in terms of FID, as expected. This effect can easily be observed qualitatively in Figure 4, where SRT and OSRT render novel views that are blurry (because they average out uncertainty about the scene), while DORSal synthesizes novel-views much more precisely by 'imagining' some of the details, while staying close to the actual content in the scene. Notabaly, in terms of LPIPS, DORSal performs the best out of all methods that condition on object representations (and thus have the same capacity for describing the content of the scene). We compare to 3DiM, which also leverages a diffusion probabilistic model, in Table 2, where we adjust DORSal to use 256 steps of DDPM (Ho et al., 2020) sampling, similar to 3DiM. It can be seen how DORSal strictly outperforms 3DiM across all metrics. Especially on Street View, where there exist large gaps between different views, 3DiM struggles to capture the content of the target view (indicated by substantially lower PSNR and higher LPIPS) as it only receives a single conditioning view during training, and primarily generates variations on its input view. We provide an additional comparison to 3DiM having access to additional GT input views at inference time in Appendix C.

## 5.2 EVALUATION OF OBJECT-LEVEL EDITS

**Setup.** We evaluate the scene editing capabilities of DORSal on both MultiShapeNet and Street View and compare to the base model, OSRT, which serves as the upper bound in our comparison. To remove objects from the scene and compute scene edit segmentation masks we follow the protocol described in Section 3.2. We compare the edit segmentation masks obtained in this way to the ground-truth instance segmentation masks for these scenes using ARI (Rand, 1971) and mIoU, which are standard metrics from the segmentation literature. As is common practice, we compute these metrics solely for foreground objects (indicated as FG-). As ground-truth instance segmentations are unavailable for Street View we only report qualitative results.

**Results.** We find that scene editing capabilities of the base OSRT model transfer to a large degree to the object-conditioned diffusion model (DORSal), even though DORSal is not trained with object-centric architectural priors or segmentation supervision. Table 3 provides a summary of quantitative scene editing results. In our comparison Sup-OSRT (Prabhudesai et al., 2023) refers to the OSRT base model trained with segmentation supervision, which provides the object slots for DORSal, i.e. this model serves as an upper bound in terms of scene editing performance (with significantly reduced visual fidelity).

Table 3: **Scene editing**. Evaluation on MultiShapeNet (metrics in %).

| Model | FG-mIoU | FG-ARI |
|---|---|---|
| OSRT | 43.1 | 79.6 |
| Sup-OSRT | 50.0 | 75.5 |
| DORSal | 45.8 | 70.0 |

On the real-world Street View dataset, the notion of an object is much more ambiguous and, unlike for MultiShapeNet, the Object Slots provided by the OSRT encoder capture individual objects less frequently. Nonetheless, we qualitatively observe how removal of individual Object Slots in DORSal can often still result in meaningful scene edits. We show a selection of successful scene edits in Figure 6, where dropping a specific slot results in the removal of, for example, a car, a street sign, a trash can, or in the alteration of a building. We provide exhaustive editing examples (incl. failure cases) in Appendix C.

We further find that slots can be transferred between scenes, with global effects such as scene lighting and object shadows correctly modeled for transferred objects. We perform slot transfer experiments by generating a single combined scene from two separate original scenes. The combined scene is obtained by taking half of the slots (i.e. latent object representations) from Scene 1 and half of the slots of Scene 2 as conditioning information for DORSal. Consequently, DORSal produces a novel scene where some objects (incl. the background) are carried over from Scene 1, mixed with objects from Scene 2. Qualitative results are shown in Figure 5.

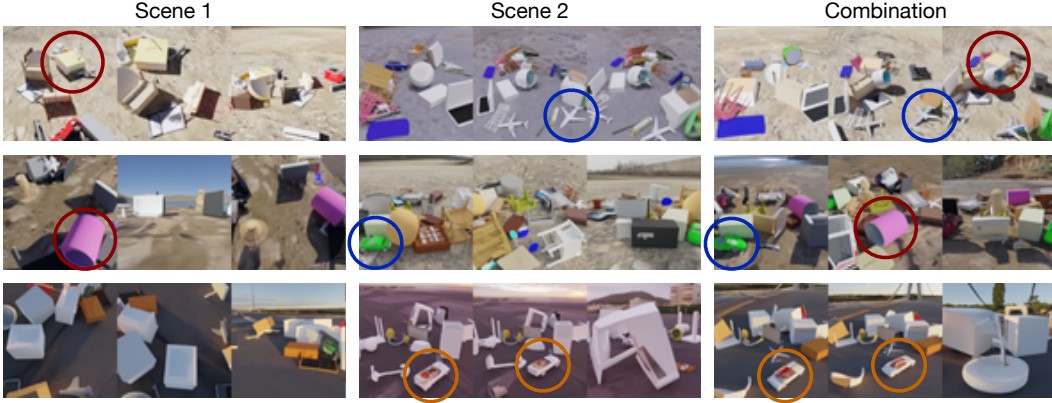

Figure 5: **Scene editing: object transfer.** We highlight several transferred objects. Note that transferred objects are rendered consistently across views (see circled objects in final row) while taking into account global illumination properties of the scene in which they are placed in (e.g. shadows are rendered correctly for transferred objects).

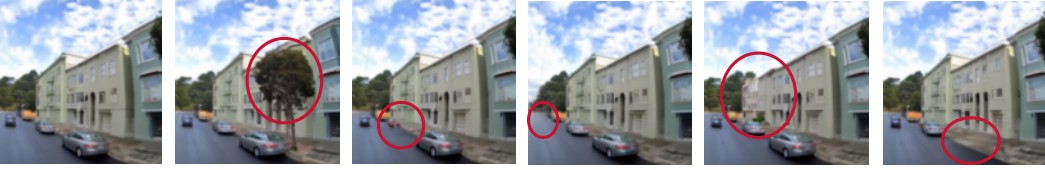

Figure 6: **Scene editing: object removal.** Removing one slot at a time, we show examples on the Street View dataset where objects are erased from the scene. Notably, the encircled tree is generated upon *removal* of a slot to fill up the now-unobserved facade previously explained by this slot. The original scene does not contain a tree in this position.

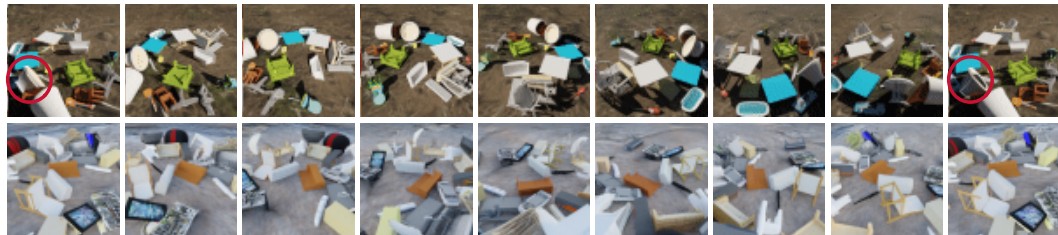

Figure 7: **Camera path rendering.** *Top*: Example of a circular camera path rendered for DORSal (64x64) trained on MultiShapeNet. While the rendered views are mostly consistent, there can be small inconsistencies in regions of high uncertainty which result in flickering artifacts (see object highlighted in red circle). *Bottom*: When trained on a dataset with close-by and fully-random camera views, DORSal achieves improved consistency resulting in qualitatively smooth videos.

## 5.3 CAMERA-PATH RENDERING

**Setup.** We qualitatively compare two different training strategies: the first is our default setup on MultiShapeNet where we train on randomly sampled views of the scene. Further, we generate a dataset which has a mix of both nearby views (w.r.t. previously generated views) and uniformly sampled views (at random) from the full view distribution. At inference time, we generate a full circular camera path for each scene using our sampling strategy described in Section 3.2.

**Results.** We show qualitative results in Figure 7 and in video format in the supplementary material. We find that DORSal is able to render certain objects which are well-represented in the scene representation (e.g. clearly visible in the input views) consistent and smoothly across a camera path, but several regions and objects "flicker" between views as the model fills in slightly different details depending on the view point to account for missing information. We find that this can be largely resolved by training DORSal on the mixed-views dataset (both nearby and random views) as described above, which results in qualitatively smooth videos. This is also reflected in our quantitative results (computed on 40 held-out scenes having 190 target views each) using PSNR as an approximate measure of scene consistency, where we obtain 16.50db PSNR for DORSal, 17.47db for 3DiM and 18.06db for DORSal trained on mixed views.

## 6 CONCLUSION

We have introduced DORSal, a generative model capable of rendering precise novel views of diverse 3D scenes. By conditioning on an object-centric scene representation, DORSal further supports scene editing: the presence of an object can be controlled by its respective object slot in the scene representation. DORSal adapts an existing text-to-video generative model architecture (Ho et al., 2022c) to controllable 3D scene generation by conditioning on camera poses and object-centric scene representations, and by training on large-scale 3D scene datasets. As we base our model on a state-of-the-art text-to-video model, this likely enables the transfer of future improvements in this model class to the task of compositional 3D scene generation, and opens the door for joint training on large-scale video and 3D scene data.

ACKNOWLEDGMENTS

We would like to thank Alexey Dosovitskiy for general advice and detailed feedback on an early version of this paper. We are grateful to Daniel Watson for making the 3DiM codebase readily available for comparison, and help with debugging and onboarding new datasets.

ETHICS STATEMENT

DORSal enables precise 3D rendering of novel views conditioned on Object Slots, as well as basic object-level editing. Though we present initial results on Street View, the practical usefulness of DORSal is still limited and thus we foresee no immediate impact on society more broadly. In the longer term, we expect that slot conditioning may facilitate greater interpretability and controllability of diffusion models. However, though we do not rely on web-scraped image-text pairs for conditioning, our approach remains susceptible to dataset selection bias (and related biases). Better understanding the extent to which these biases affect model performance (and interpretability) will be important for mitigating future negative societal impacts that could arise from this line of work.

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

## A    LIMITATIONS

While DORSal makes significant progress, there are several limitations and open problems worth highlighting, relating to 1) lack of end-to-end training, 2) worse editing performance and consistency for high-resolution training, 3) configuration of the MultiView U-Net architecture for 3D, 4) non-local editing effects, and 5) .

As we follow the design of Video Diffusion Models (Ho et al., 2022c) for simplicity, DORSal is not end-to-end trained and is ultimately limited by the quality of the scene representation (Object Slots) provided by the separately trained upstream model (OSRT). End-to-end training comes with additional challenges (e.g. higher memory requirements), but is worth exploring in future work.

We further found that training at 128x128 resolution with our model design results in decreased editing performance compared to a 64x64 model. We also observed qualitatively worse cross-view consistency in the higher-resolution model. To overcome this limitation, one would likely have to scale the model further in terms of size (at the expense of increased compute and memory requirements) or train a cascade of models to initially predict at 64x64 resolution, followed by one or more conditional upsampling stages, as done in Video Diffusion Models (Ho et al., 2022c).

As the U-Net architecture of DORSal is based on Video Diffusion Models (Ho et al., 2022c), it can be sensitive to ordering of frames in the dataset. While frames in MultiShapeNet are generated from random view points, frames in Street View are ordered by time. DORSal is able to capture this information, which—in turn—makes rendering views from arbitrarily chosen camera paths at test time challenging, as the model has learned a prior for the movement of the camera in the dataset.

For scene editing, we find that removing individual object slots can have non-local side effects, e.g. another object or the background changing its appearance, in some cases. Furthermore, edits of individual are typically not perfect, even when trained with a supervised OSRT base model: objects are sometimes only partially removed, or removal of a slot might have no effect at all. Especially on Street View, not all edits are meaningful and many slots have little to no effect when removed, likely because the OSRT base model often assigns multiple slots to a single object such as a car. This qualitative result, however, remains remarkable as the base OSRT model received no instance supervision whatsoever.

Finally, we would like to point out how the scene editing operations that are currently supported (eg. object removal, transfer) are limited and supporting more fine-grained scene editing operations is an important direction for future work. For object-level edits, such as applying a rotation or translation, it is foreseeable how supervised co-training with language can provide an interface for this as in Michel et al. (2023). Alternatively, the object representations themselves could be disentangled to a point where information about the rotation or position of an object is isolated, and can thus be manipulated independently during generation (Biza et al., 2023; Singh et al., 2022).

## B    EXPERIMENTAL DETAILS

### B.1    EVALUATION

**Novel View Synthesis.**  We follow the experimentation protocol outlined in Sajjadi et al. (2022a;c) and evaluate DORSal and baselines using 5 and 3 novel target views for MultiShapeNet and Street View respectively. Similarly, the OSRT base model, is trained with 5 input views on these datasets. To accommodate the U-Net architecture used in DORSal and 3DiM, we crop Street View frames to 128x128 resolution.

**Evaluation of Object-level Edits.**  We use a median kernel size of 7 for all edit evaluations (incl. the baselines). We evaluate models on the first 1k scenes of the MultiShapeNet dataset. For DORSal, we use an identical initial noise variable (image) for each edit to ensure consistency. We use the 64x64 DORSal architecture for this set of experiment to allow for faster sampling of all possible object edits and since we found that the lower-resolution model is less susceptible to side effects during editing (e.g. slight changes in other parts of the scene), which result in a lower edit scores for the 128x128 model (60.6 vs. 70 FG-mIoU). This suggests that a good strategy for optimal object-level control of scene content would be to train a model at 64x64 resolution followed by one or more upsampling stages (Saharia et al., 2022c; Ho et al., 2022a).

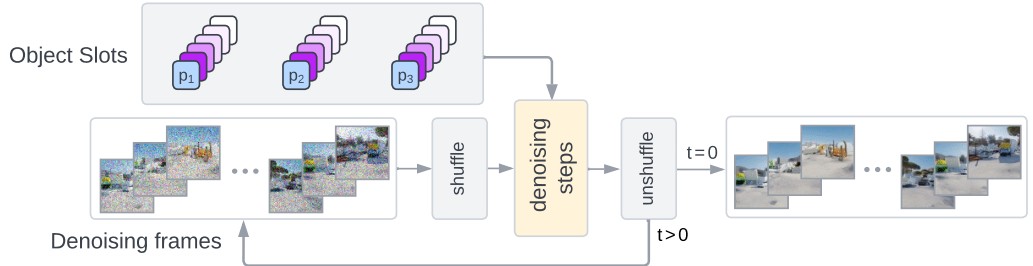

Figure 8: **View consistent rendering for a large number of frames.** Frames are denoised in blocks of length $L$ (e.g. $L = 3, 5$ or 10). To ensure consistency between blocks, the shuffle component acts in one of the following three ways: 1) identity (do nothing) 2) shift the frames by about half the context length, for smoothness between neighbouring blocks, and 3) a random permutation for global consistency.

**Camera-path Rendering.** For camera-path rendering of many views (beyond what DORSal was trained with) we deploy the sampling procedure outlined in Section 5.3. Camera trajectories follow a circular, center-facing path starting from the first input view. Further, we generate a dataset which has a mix of both nearby views (w.r.t. previously generated views) and uniformly sampled views (at random) from the full view distribution as in MultiShapeNet. Here we train DORSal using 10 input views and 10 target views (down-sampled to 64x64 resolution) to keep a similar amount of diversity when sampling far-away as well as close-by views. 3DiM is trained similarly as for the novel-view synthesis experiments.

## B.2 MODEL DETAILS

### B.2.1 DORSAL

**Conditioning.** We obtain Object Slots from a separately trained OSRT model. In the case of MultiShapeNet, we train OSRT with instance segmentation mask supervision following the approach by Prabhudesai et al. (2023): we take the alpha masks produced by the broadcast decoder to obtain soft segmentation masks, which we match using Hungarian matching with ground-truth instance masks (under an L2 objective) and finally train the model using a cross-entropy loss using the alpha mask logits on the matched target masks. For Street View, we use the default unsupervised OSRT model with a broadcast decoder, as instance masks are not available. All OSRT models use 32 Object Slots.

**Slot Dropout.** Ideally, slot representations that summarize the scene should be conditionally independent given an image $z_t$, $p(s_{1:K}|z_t) = \prod_{k=1,K} p(s_i|z_t)$, i.e. to be able to manipulate the presence of objects independently for editing purposes. In reality, the slot representations may respect this assumption to varying degrees, with an OSRT model trained with instance-level supervision (Sup-OSRT) being more likely to achieve this. However, even if slots would exclusively bind to particular regions of the encoded input views that correspond to individual objects, slots may still share information as the input view encoder has a global receptive field. To mitigate this issue, we experimented with dropping slots from the conditioning set independently following a Bernoulli rate set as a hyper-parameter $\lambda_{sd}$. In this case the model sees slot subsets at training time (such that edits are now effectively in-distribution). While we found that this slightly affected Edit FG-ARI results for MultiShapeNet in a negative way, we found that it qualitatively resulted in more consistent object edits on Street View. See Appendix C.1 for a comparison. Unless otherwise mentioned we report results using $\lambda_{sd} = 0$ for MultiShapeNet and $\lambda_{sd} = 0.2$ for Street View.

**Network Architecture.** For the DORSal we follow the architecture of Ho et al. (2022a), which is a U-Net that has axial attention over time, whose specification is as follows:

- The inputs are the noisy set of $L$ target views (acting as the noisy video). The inputs are processed at multiple attention resolutions, each corresponding to a "level", followed by a spatial down-sampling by a factor of two. Each level in the downward path is composed of three ResNet blocks

Table 4: DORSal U-Net architecture details.

| Model | Channels per level | Blocks per level | Attention resolution | Patching |
|-------|--------------------|------------------|----------------------|----------|
| U-Net 64 | 192, 384, 576 | 3 | $16 \times 16$ | No |
| U-Net 128 | 256, 512, 1024 | 3 | $16 \times 16$ | $2 \times 2$ |

having the amount of channels as indicated in Table 4. The middle stage consists of a single ResNet block (keeping the number of channels constant). The upward path mimics the downward path in reverse and has a residual connection to the corresponding block in the downward path. Attention takes place only at the third level (spatial resolution 16x16 after each of the ResNet blocks) in the downsample, middle and upsample paths, using a head dimensionality of 64 and 128 for input resolution 64x64 and 128x128 respectively.

- The UNet is further conditioned with embeddings of the slots, target pose and diffusion noise level. The individual object slots are projected and broadcasted across views, where they are combined with the target camera pose (after projection) for each of the L views. We use sinusoidal pose embeddings of absolute camera rays, identical to the setup in OSRT (Sajjadi et al., 2022a). We apply this conditioning in the same way that text is integrated into the U-net: we attention-pool the conditioning embeddings into a single embedding and combine it with the embedding for the diffusion noise level for modulating U-Net feature maps via FiLM (Perez et al., 2018). Further, we use cross-attention as indicated above to attend to the conditioning embeddings derived from the object slots and camera poses.

A key difference is that DORSal does not require text conditioning cross attention layers, and it solely uses slot embeddings augmented with camera poses as indicated above. Further, notice how the architecture sizes we use are small compared to Ho et al. (2022a) as can be seen in Table 4. The U-Net on resolutions of $128 \times 128$ uses patching to avoid memory expensive feature maps. For the $16 \times 16$ resolution, the ResBlocks use per-view self-attention and between-views cross-attention.

For details about the OSRT encoder used to compute the frozen object representations, we refer to Section B.2.3 below.

**Training.** We adopt a similar training set-up to Ho et al. (2022c), using a cosine-shaped noise schedule, a learning rate with a peak value of 0.00003 using linear warm-up for 5000 steps, optimization using Adam with $\beta_1 = 0.9$, $\beta_2 = 0.999$, and EMA decay for the model parameters. We train with a global batch size of 8 and classifier-free guidance with a conditioning dropout probability of 0.1 (and an inference guidance weight of 2). We report results after training for $1\,000\,000$ steps. For MultiShapeNet, we use Object Slots from Sup-OSRT (i.e. supervised) and do not use slot dropout during training. For Street View, we use Object Slots from OSRT (i.e. unsupervised) and use a slot dropout probability of 0.2, which we found to improve editing quality on this dataset (compared to no slot dropout).

**Camera-Path Sampling.** We leverage the iterative nature of the generative denoising process to create *smooth transitions* as well as *global consistency* between frames. Our technique is summarized in Figure 8: we iterate over a total of 25 *stages* (i.e. 8 steps per stage when using 200 denoising steps) and we interleave 3 types of shuffling (1 type per stage) to achieve both local and global consistency. The types of shuffling (as described in Section 3.2) are as follows: 1) no shuffle (identity), to allow the model to make blocks of the context length consistent; 2) shift the frames in time by about half of the context length, which puts frames together with new neighbours in their context, allowing the model to create smooth transitions; 3) shuffle all frames with a random permutation, to allow the model to resolve inconsistencies globally.

### B.2.2 3D DIFFUSION MODEL (3DIM)

We compare to 3DiM, which is a pose-conditional image-to-image diffusion model for generating novel views of the same scene (Watson et al., 2023). During training, 3DiM takes as input a pair of views of a static scene (including their poses), where one of the views (designated as the "target view") is corrupted with noise. The training objective is to predict the Gaussian noise that was used to corrupt the target view. During inference, 3DiM makes use of *stochastic conditioning* to generate 3D-consistent views of a scene. In particular, given a small set of $k$ conditioning views

and their camera poses (typically $k = 1$), a new view for a given target camera pose is generated by conditioning on a randomly selected view from the conditioning set at each denoising step. Each time a new view is generated, it is added to the conditioning set. For additional details, including code, we refer to Sections 6 & 7 in Watson et al. (2023).

**Network Architecture.** In our experiments we use the default $\sim$471M parameter version of their X-UNet, which amounts to a base channel dimension of $ch = 256$, four stages for down- and up-sampling using $ch\_mult = (1, 2, 2, 4)$, and 3 ResBlocks per stage using per-view self-attention and between-views cross-attention at resolutions $(8, 16, 32)$. Note how this configuration uses many more parameters per view, compared to DORSal. In line with DORSal, we use absolute positional encodings for the camera rays in our experiments on MultiShapeNet and StreetView (scaling down the ray origins by a factor of 30).

**Training.** We adopt the same training set-up as in the 3DiM paper, which consist of a cosine-shaped noise schedule, a learning rate with peak value of 0.0001 using linear warm-up for 10M samples, optimization using Adam with $\beta_1 = 0.9$ and $\beta_2 = 0.99$, and EMA decay for the model parameters. We train with a batch size of 128 and classifier-free guidance $10\%$ with a weight of 3, as was done for the experiment on SRN cars in their paper. We report results after training for $320\,000$ steps.

**Sampling.** We generate samples in the same way as in the 3DiM paper, using 256 DDPM denoising steps and clip to [-1, 1] after each step.

### B.2.3 SRT & OSRT

SRT was originally proposed by Sajjadi et al. (2022c) with Set-Latent Scene Representations (SLSR) and subsequently adapted to Object Slots for OSRT (Sajjadi et al., 2022a). At the same time, a few tweaks were made to the model, e.g. by using a smaller patch size and a larger render MLP (Sajjadi et al., 2022a). For all our experiments (SRT and OSRT), we use the improved architecture from the OSRT paper. We reproduce several key details for the encoder, which is used to compute the object representations for DORSal, and refer to Appendix A.4 in (Sajjadi et al., 2022a) for additional model and training details.

- The encoder consists of a CNN with 3 blocks, each with 2 convolutional layers and ReLU activations. The first convolution in each block has stride 1, the second has stride 2. It begins with 96 channels, which are doubled with every strided convolution. The final activations are mapped with a 1x1 convolution (i.e. a per-patch linear layer) to 768 channels.

- The CNN is followed by an encoder transformer, using 5 pre-normalization layers with self-attention (Xiong et al., 2020). Each layer has hidden size 768 (12 heads, each with 64 channels), and the MLPs have 1 hidden layer with 1536 channels and GELU activations (Hendrycks & Gimpel, 2016).

- The encoder transformer is followed by a Slot Attention module (Locatello et al., 2020) using 1536 dimensions for slots and embeddings in the attention layers. The MLP doubles the feature size in the hidden layer to 3072. We use a single iteration of Slot Attention with 32 slots.

We use identical encoder architectures between OSRT and DORSal. Following Sajjadi et al. (2022c), we seperately train SRT and OSRT for $\sim$4M steps for each dataset.

### B.3 Compute and Data Licenses

We train DORSal on 8 TPU v4 (Jouppi et al., 2023) chips using a batch size of 8 for approx. one week to reach 1M steps. The MultiShapeNet dataset was introduced by Sajjadi et al. (2022c) and was generated using Kubric (Greff et al., 2022), which is available under an Apache 2.0 license. Street View imagery and permission for publication have been obtained from the authors (Google, 2007).

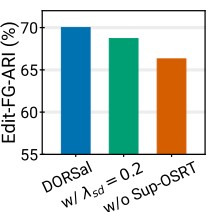 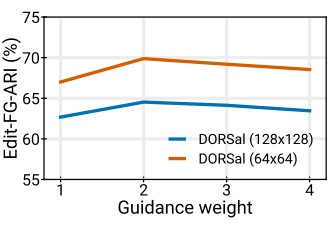 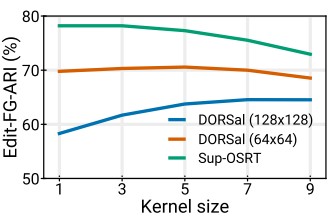

(a) Slot dropout & base model.      (b) Guidance weight.      (c) Median filter kernel size.

Figure 9: **Hyperparameter choices and ablations**. In (a), we compare DORSal (64x64) without slot dropout ($\lambda_{sd} = 0$) with two variants, $\lambda_{sd} = 0.2$ and using an unsupervised OSRT model (w/o Sup-OSRT) as base. In (b), we analyse the effect of the guidance weight parameter during inference, and in (c) we show the effect of kernel size on the median filter used during scene edit evaluation.

## C    ADDITIONAL RESULTS

### C.1    ABLATIONS

We investigate the effect of 1) slot dropout, 2) instance segmentation supervision in the base OSRT model (for MultiShapeNet), 3) the guidance weight during inference, and 4) the median filter kernel size for scene edit evaluation. Our results are summarized in Figure 9.

We find that adding slot dropout can have a negative effect on scene editing metrics in Multi-ShapeNet for which we use Sup-OSRT as the base model (Figure 9a). This is interesting, since for Street View, where supervision is not available, we generally report results using a model with $\lambda_{sd} = 0.2$, as the model without slot dropout did not produce meaningful scene edits. Removing instance supervision in in MultiShapeNet in the OSRT base model expectedly reduces scene editing performance (Figure 9a). Further, we find that choosing a guidance weight larger than 1 generally as a positive effect on prediction quality, with an optimal value of 2 (Figure 9b).

An important hyperparameter for scene editing evaluation is the median filter kernel size, which sets an upper bound on achievable segmentation performance (as fine-grained details are lost), yet is important for removing sensitivity to high-frequency details which can often vary between multiple samples in a generative model. We find that DORSal at 128x128 resolution benefits from smoothing up to a kernel size of 7 (our chosen default), which slightly lowers the achievable segmentation score of the base model (Sup-OSRT), but removes most noise artifacts in our edit evaluation protocol (Figure 9c).

### C.2    VARIANCE

While we report results for a single representative model run in the main paper, we found that variance in quantitative metrics across different runs is fairly moderate. Specifically, we measure a standard error of approx. 0.004 for LPIPS and approx. 0.2 for PSNR (for 3 model training re-runs with different initialization seeds), which does not affect the interpretation of our reported results.

### C.3    3D CONSISTENCY

To provide some further insight into the 3D consistency of DORSal, we re-computed the edit-segmentation scores on a subset of the samples where we measure both Edit FG-ARI across all the views simultaneously and a "2D Edit FG-ARI" where we compute Edit FG-ARI for each view individually and then average the results. Note that the latter does not penalize inconsistencies between the views, hence the gap between the two scores is indicative of any inconsistencies taking place. A similar approach to evaluating 3D consistency was carried out in Sajjadi et al. (2022a). We obtain 0.702 Edit FG-ARI and 0.721 2D Edit FG-ARI. The small gap between these scores indicates that the segmentation obtained via this procedure is highly consistent across views.

### C.4    REPRODUCING INPUT VIEWS

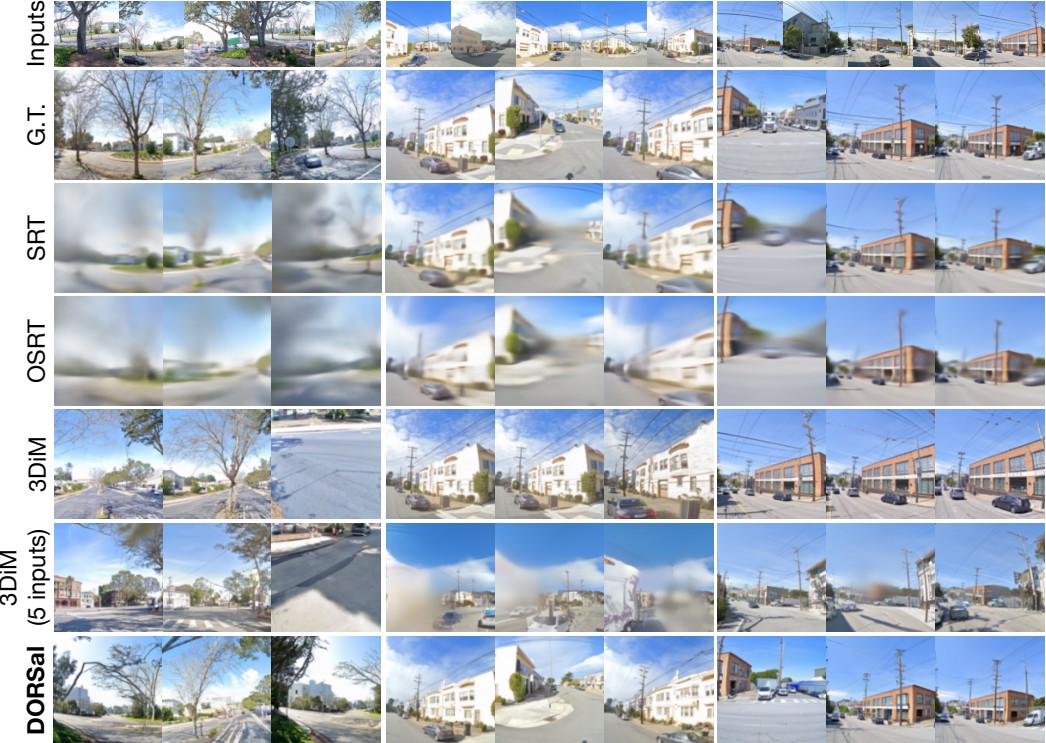

Figure 10: **Novel View Synthesis (Street View)**. Qualitative results incl. input views (top tow) for additional Street View scenes. We further include a version of 3DiM that is conditioned on 5 ground-truth input views.

To give an indication of whether the slots contain sufficient information about the scene, we ran DORSal and sup-OSRT on 16 scenes of MultiShapeNet with target views matching input views (Table 5). For both models, we observe how predicting input views improves PSNR and LPIPS (compared to their respective novel-view synthesis scores in Table 1), indicating that the models are better able to leverage the information contained in object slots in this setting. In particular, the significantly lower LPIPS score for DORSal indicates faithful reconstruction of the inputs. This is a positive result for DORSal in particular, suggesting that it hallucinates less in these instances.

Table 5: **Reproducing input views**. Eval on 16 MultiShapeNet scenes.

| Model | PSNR | LPIPS |
|---|---|---|
| Sup-OSRT | 23.72 | 0.340 |
| DORSal | 19.55 | 0.252 |

### C.5 QUALITATIVE RESULTS

**Novel View Synthesis** We provide additional qualitative novel view synthesis results in Figures 10 (Street View) and 11 (MultiShapeNet). For Street View, it is evident that even when modifying 3DiM to use 5 ground-truth input views during inference, it is unable to synthesize accurate views from novel directions, while DORSal renders realistic views that adhere to the content of the scene.

**Scene Editing** We show qualitative results for scene editing in MultiShapeNet in Figure 12 and for Street View in Figure 13. In Figure 14 we further provide exhaustive scene editing results for several Street View scenes: each image shows one generation of DORSal with exactly one slot removed. These results further highlight that several meaningful edits can be made per scene. Typical failure modes can also be observed: 1) some objects are unaffected by slot removal, 2) some edits have side effects (e.g. another object disappearing or changing its appearance), and 3) multiple different edits have the same (or a very similar) effect. These failure modes likely originate in part from the unsupervised nature of the OSRT base model, which sometimes assigns multiple slots to a single

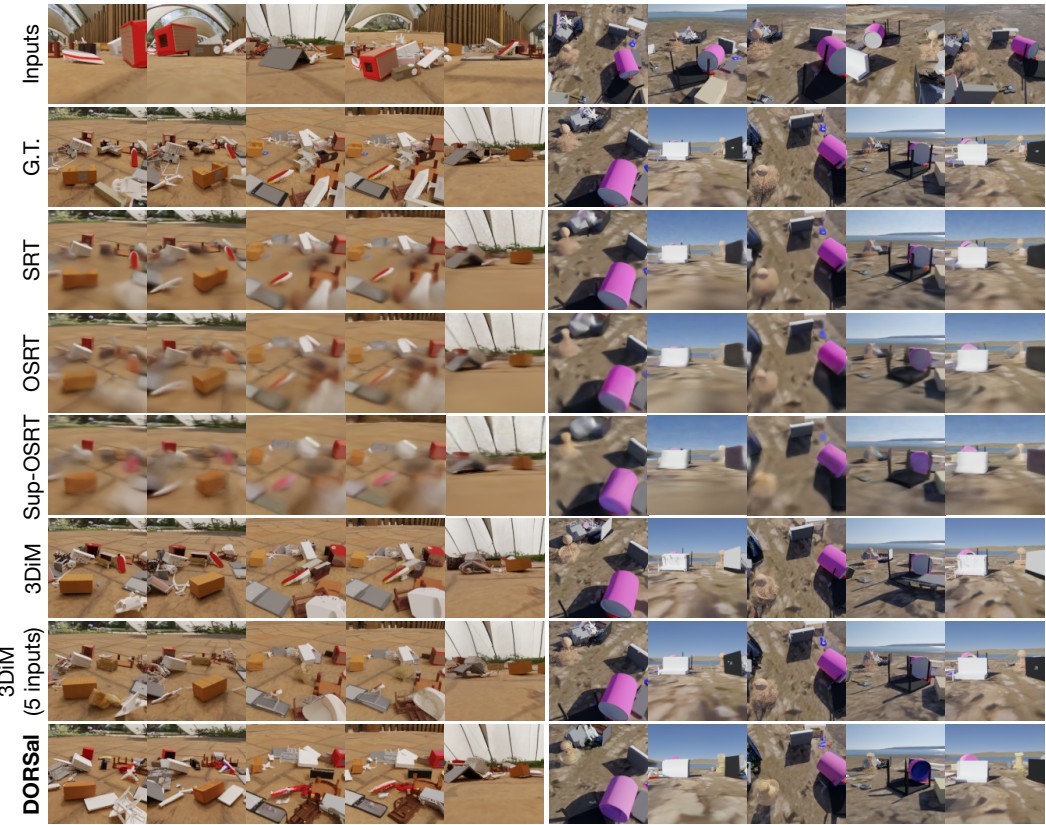

Figure 11: **Novel View Synthesis (MultiShapeNet)**. Qualitative results incl. input views (top tow) for additional MultiShapeNet scenes. We further include Sup-OSRT, which is trained using segmentation supervision (and provides the Object Slots for DORSal on MultiShapeNet), and a version of 3DiM that is conditioned on 5 ground-truth input views.

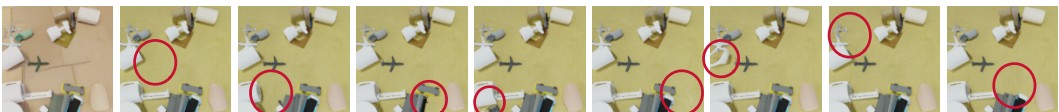

Figure 12: **Scene Editing (MultiShapeNet).** We remove one slot at a time in the conditioning of DORSal and render the resulting scene while keeping the initial image noise fixed. In the leftmost panel, the slot corresponding to the background is removed while all objects are present. The other panels show deleted objects (highlighted in red circles) when their corresponding slot is removed.

object, or does not decompose the scene well. Fully "imagined" objects (i.e. objects which are not visible in the input views and therefore not encoded in the Object Slots) further generally cannot be directly edited in this way. Some of these issues can likely be overcome in future work by incorporating object supervision (as done for MultiShapeNet), and by devising a mechanism by which "imagined" objects not visible in input views are similarly encoded in Object Slots.

## C.6 COMPARISON TO 3DiM USING ADDITIONAL INPUT VIEWS

The stochastic conditioning procedure used during sampling from 3DiM can be initialized with an arbitrary number of ground-truth input views. In the main paper, we follow the implementation details from Watson et al. (2023) and use a single ground-truth input view. However, because DORSal conditions on Object Slots computed from five input views, it would be informative to increase the number of input views to initialize 3DiM sampling accordingly. The results for this experiment are reported in Table 6, where it can be seen how 3DiM performs markedly better on MultiShapeNet

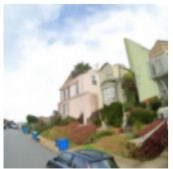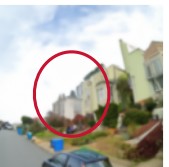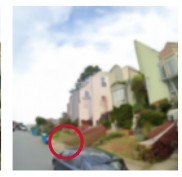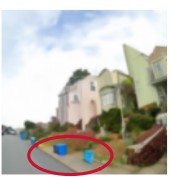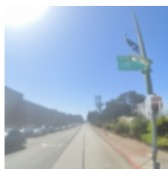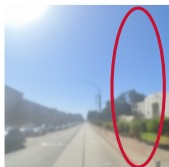

Figure 13: **Scene Editing (Stree View).** Removing one slot at a time, we here show further examples on the Street View dataset where objects are erased from the scene.

Table 6: **Novel-view synthesis**. Using additional ground-truth input views for 3DiM.

| Model | MultiShapeNet | | | StreetView | | |
|---|---|---|---|---|---|---|
| | PSNR↑ | LPIPS↓ | FID↓ | PSNR↑ | LPIPS↓ | FID↓ |
| 3DiM | 18.20 | 0.287 | 10.94 | 12.68 | 0.477 | 15.58 |
| 3DiM (5 input) | **21.46** | **0.182** | **8.20** | 12.25 | 0.557 | 34.47 |
| DORSal (DDPM) | 18.99 | 0.265 | 9.00 | **16.36** | **0.356** | **14.62** |

in this case. In contrast, on Street View the opposite effect can be seen, where 3DiM performs markedly worse in this case.

We hypothesize that this difference is due to how well 3DiM performs after training on these datasets. On MultiShapeNet, 3DiM achieves a better training loss and renders novels views that are close to the ground truth. Hence, initializing stochastic conditioning with additional views, will help provide more information about the actual content of the scene and thus help produce better samples. In contrast, 3DiM struggles to learn a good solution during training on Street View due to large gaps between cameras (and the increased complexity of the scene) and resorts to generating target views close to its input view. Hence, increasing the diversity of the ground-truth input views, will cause the model to generate views that lie in between these, which hurts its overall performance.

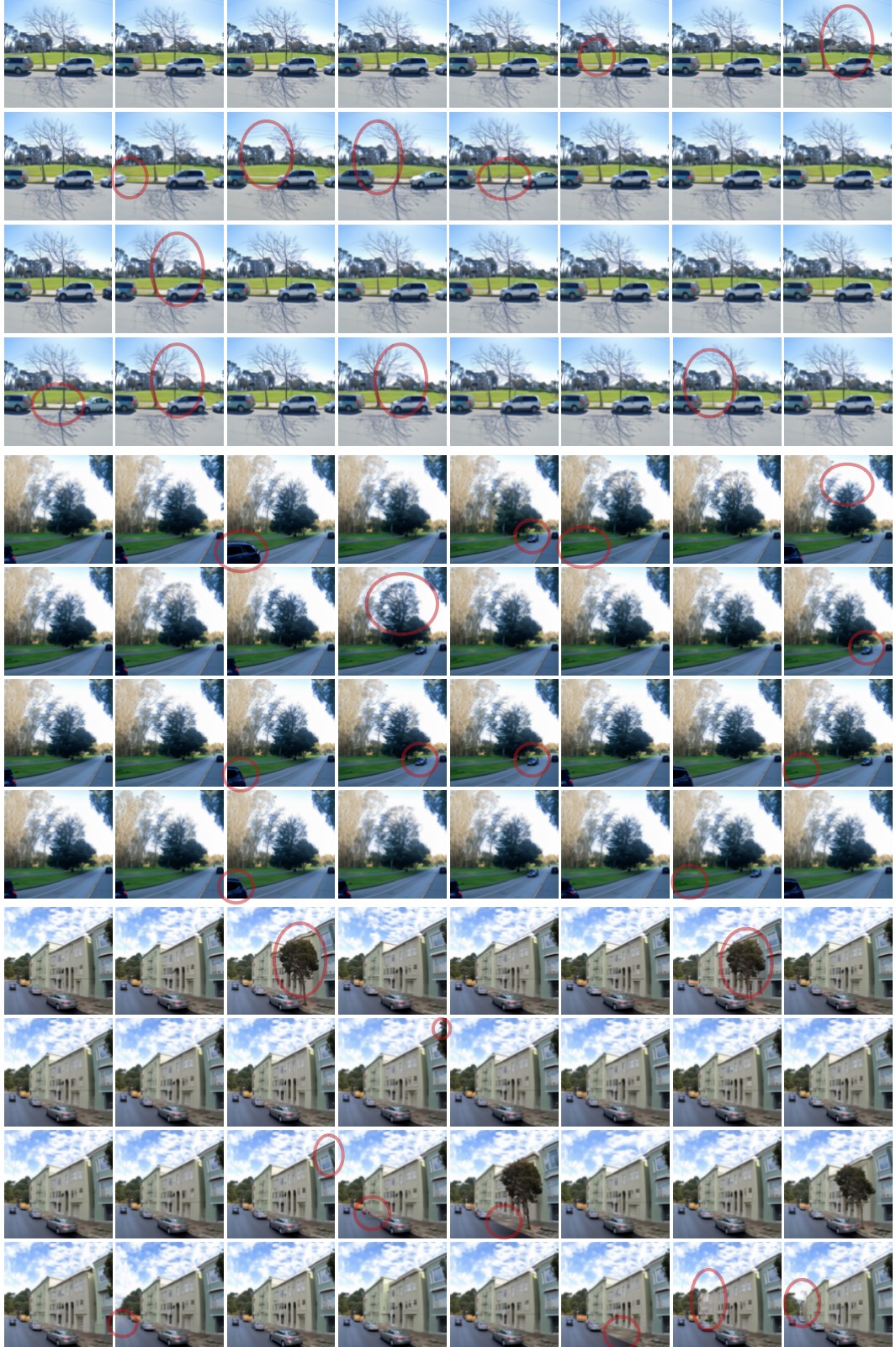

Figure 14: **Scene Editing (Street View; Exhaustive)**. Exhaustive DORSal scene editing results for three Street View scenes, with one Object Slot removed at a time. Several examples where scene content differs are highlighted.

