# OpenReview forum: "DORSal: Diffusion for Object-centric Representations of Scenes $\textit{et al.}$"
_ICLR.cc/2024/Conference — ICLR 2024 poster_

### Official Review · Reviewer_J43q · 2023-10-31

**Soundness:** 3 good
**Presentation:** 3 good
**Contribution:** 3 good
**Rating:** 6
**Confidence:** 4

**Summary:**

This paper proposes a novel approach for controllable scene synthesis with an object-centric representation. It first extracts object slots, the object-centric representation, using an auto-encoder, and then trains a multi-view diffusion model conditioned on these object slots for novel view synthesis. Extensive experiments on MultishapeNet and Street View datasets support the effectiveness of the proposed method.

**Strengths:**

1. The paper demonstrates the effectiveness of using an object-centric representation for controllable scene synthesis. The object slots can be learned in an unsupervised manner and represent the scene decompositionally. Training a multi-view diffusion model conditioned on such representation can support high-quality scene synthesis, and enable many applications like object removal and transferring.

2. This paper is well-written. It is easy-to-follow and contains much details.

**Weaknesses:**

1. The paper only shows results with low resolution. Experiments with higher resolution can further reveal the potential of the proposed method.

2. The proposed scene editting scheme can only support object removal and transferring. Can the framework be extended to support more diverse scene editing operations (like translation, rotation).

**Questions:**

1. Since you incorporate the Street View dataset, which features complex and unbounded outdoor scenes, I am particularly interested in the OSRT performance in this dataset. Could you please provide object-decomposed visualization as illustrated in the OSRT paper (Fig. 4). These visualizations would reveal how different portions of the street view are represented by individual slots.

2. What is the rationale of incorporating a video (multi-view) diffusion model for synthesis? If the capacity of OSRT is strong enough, will a single-view diffusion model (conditioned on OSRT and single view direction) be enough to generate results with good multi-view consistency?

---

> ### Author Response · Authors · 2023-11-15
> **Response to reviewer J43q (part 1)**
>
> Thank you for your review and suggestions to improve our work! We are pleased that you recognize how our extensive experiments on MultishapeNet and Street View datasets support the effectiveness of our approach, and how this research can enable many applications.
>
> In terms of the Weaknesses (W) and Questions (Q) you list, our response is as follows:
>
> **W1: The paper only shows results with low resolution.**
>
> Indeed, we agree that experiments with higher resolution would be even more impactful. However, the resolution of 128x128 we report results on is typical for relevant prior work. In fact, both the diffusion-based 3DiM baseline and the SRT-based baselines report only up to 128x128, as does the video diffusion architecture we base our decoder on.
>
> Generally speaking, high-resolution, consistent video generation (or in our case generating multiple views of the same scene) is a challenging open problem (Hoogeboom et al., “End-to-end diffusion for high resolution images”, 2023). Fortunately, a straightforward approach to increase the resolution is via cascading: having multiple stages of diffusion, where later stages perform super-resolution to arbitrarily increase the resolution (Ho et al., “Cascaded diffusion models for high fidelity image generation”, 2022). This approach is equally applicable to DORSal, though we have not considered it in practice, as high resolution generation would require an additional standalone diffusion model and is not the main focus of our work.
>
> **W2: The proposed scene editting scheme can only support object removal and transferring.**
>
> Indeed, we acknowledge that this is a limitation of our current approach and supporting more fine-grained scene editing operations is an important direction for future work. For the rotation and translation you mentioned in particular, it is foreseeable how supervised co-training with language can provide an interface for this as in “Object 3DIT” (Mochel et al., 2023), which we cite in our work. Alternatively, the object representations themselves could be disentangled to a point, where information about the rotation or position of an object is isolated, and can thus be manipulated independently during generation. In general, fine-grained, object-level control is an active area of research, and approaches that offer such capabilities like DisCoScene (Xu et al., 2023) requires a lot more prior knowledge about the scene to be able to work.
>
> We recognize that the current discussion of limitations is perhaps too lenient in this regard. In the revision, we will update it to comment on fine-grained control.
>
> **Q1: Could you please provide object-decomposed visualization as illustrated in the OSRT paper (Fig. 4).**
>
> Thank you for your interest. Figure 1 in the current submission contains a representative example of how OSRT assigned different parts of the scene to different slots (like Figure 4 in the OSRT paper). For completeness, we have uploaded the same visualization for the other target views as well, and for an additional scene, in the supplementary material (*osrt_streetview_decomposition.pdf*). It can be seen how OSRT (trained fully unsupervised here) usually offers a coarse decomposition of the background into broad segments and partially succeeds at decomposing some of the objects.
>
> We note how OSRT was never evaluated on Street View in the first place, and is clearly pushing the limits of what it is capable of. For this reason we show interesting cases in Figure 6 and exhaustive edits including failure cases in Figure 13 in Appendix C. In Appendix C.2 we comment on how “These failure modes likely originate in part from the unsupervised nature of the OSRT base model, which sometimes assigns multiple slots to a single object, or does not decompose the scene well. Fully “imagined” objects (i.e. objects which are not visible in the input views and therefore not encoded in the Object Slots) further generally cannot be directly edited in this way.”. In that same section we also hint at possible solutions for addressing this.

---

> > ### Author Response · Authors · 2023-11-15
> > **Response to reviewer J43q (part 2)**
> >
> > **Q2:  What is the rationale of incorporating a video (multi-view) diffusion model for synthesis? If the capacity of OSRT is strong enough, will a single-view diffusion model (conditioned on OSRT and single view direction) be enough to generate results with good multi-view consistency?**
> >
> > It is certainly an interesting question whether a single view direction can yield plausible generations by conditioning on OSRT representations, though it would be impossible for the model to synchronize newly added information across views, unless one follows a different strategy for including information from other related views. This is also evident from related work, such as 3DiM, where “stochastic conditioning” (i.e. conditioning the denoising process on a different image from the conditioning set at each step) during inference time is critical for multiview consistency.
> >
> > Another benefit of multi-view generation is the interaction with the shuffling techniques for camera-path rendering explored in our paper. Basically, the larger the size of the context window of each view (i.e. the total number of views that are generated simultaneously and can thus interact with one another to resolve inconsistencies), the fewer shuffling operations we need to synchronize all of the views when sampling camera trajectories for many steps.

---

> > > ### Comment · Reviewer_J43q · 2023-11-22
> > >
> > > Hi, authors. Apologies for my delayed response as I just recovered from the deadlines of ICLR and CVPR. Thank you for your reply, which has addressed most of my concerns. However, I still have some reservations regarding the editing capacity.
> > >
> > > You mentioned that "it is foreseeable how supervised co-training with language can provide an interface for this as in 'Object 3DIT' (Mochel et al., 2023)." However, Object 3DIT relies on a paired dataset for supervised training with language. I am uncertain how this technique can be applied to more general cases without language captions, such as the street view dataset you used. Additionally, you mentioned that "the object representations themselves could be disentangled to a point, where information about the rotation or position of an object is isolated, and can thus be manipulated independently during generation." I am also unclear about how to achieve this kind of disentanglement based on the current method without significant modification.

---

> > > > ### Author Response · Authors · 2023-11-22
> > > > **Regarding editing capacity.**
> > > >
> > > > Thank you so much for getting back to us. Below we hope to clear up some ways in which DORSal can enable fine-grained object-level scene control in the future with minimal changes.
> > > >
> > > > Concretely, one straightforward approach for obtaining control over object placement, scale, and possibly rotation is to use Invariant Slot Attention (ISA, Biza et al., 2023). ISA works by separating information about the position and scale of an object from the Object slot, such that its object representation becomes invariant to these transformations. If we train OSRT with ISA as a drop-in placement for Slot Attention, then we obtain a set of object slots (as we have now), and further for each slot the position and scale of each object in the scene. We can condition the Diffusion decoder in DORSal on this extra positional / scale information in the same way that we condition on target camera position. Hence, in all likelihood, this should provide us with object-level control during generation and let us change the position / scale of each object independently. We note that the approach outlined above is fully unsupervised, and therefore equally applicable to StreetView or any other datasets, while requiring minimal changes.
> > > >
> > > > Another approach is to assume access to a small amount of annotations, such as pairs of scenes s, s' that differ only by a local transformation to one object. These annotations are straightforward to obtain for synthetic data, and could be used to train the diffusion decoder to pay attention to an extra (object-level) latent variable that encodes information about the transformation that was applied. As a concrete example, we could extract the object slots from one scene s and train the diffusion decoder to generate views for another scene s’ conditioned on these object slots + an extra bit of information about the transformation that was applied between the scenes. Sticking with the Object 3DIT setting, language might be one way with which to encode information about this transformation, though other, more easily applicable, set-ups are possible too (i.e. a simple indicator for the object to transform and x,y,z variables to specify the offset for a change in position). In all likelihood, by additionally training the diffusion decoder in this way, this should provide for fine-grained control. Further, co-training with synthetic data might make it possible to benefit from such control in scenes where no annotations are available.
> > > >
> > > > More generally, we would like to emphasize that the contribution of this paper is twofold. Although one part of the contribution of this work is to open up the possibility for scene editing by conditioning on object representations (something that prior NVS + diffusion approaches simply can not provide for), the arguably currently larger contribution of this work is to make progress towards NVS from few observations in real-world multi-object scenes. To that end, we demonstrate that conditioning a video diffusion architecture on (structured) scene representations and pose encodings allows us to overcome limitations of a prior state-of-the-art 3D generative diffusion model (3DiM).  In demonstrating the feasibility of this approach (and what’s more, how well it actually fares against the baselines), we expect follow-up work to be able to make further progress toward achieving the ambitious goals of real-world NVS and object-level control.
> > > >
> > > > Biza et al. (2023), “Invariant Slot Attention: Object Discovery with Slot-Centric Reference Frames”, ICML 2023, https://arxiv.org/abs/2302.04973

---

### Official Review · Reviewer_dJMA · 2023-10-31

**Soundness:** 4 excellent
**Presentation:** 3 good
**Contribution:** 3 good
**Rating:** 6
**Confidence:** 4

**Summary:**

The paper extends the Object Scene Representation Transformer (OSRT) with a diffusion-based decoder, proposing DORSal.
It enables a model to render precise images while maintaining properties of OSRT, i.e., (unsupervised) decomposition of objects.
The experiments were conducted with appropriate baselines, metrics, and datasets.

**Strengths:**

- Straightforward success of the proposal idea to an important problem, i.e., diffusion decoder for OSRT-based NVS. It enables the models to generate sharp images with object-centric properties. This is a nice contribution to the research of SRT because the SRT family has been likely to render very blurred images, which is one of the largest weaknesses.
- Easy to read and well-written manuscript. The text is very fluent and informative.

**Weaknesses:**

- A little reserved technical novelty. The usage and its effectiveness of diffusion-based decoders for precise rendering of NVS models are already widely spread this year (Watson et al., 2023; Chan et al., 2023; Tewari et al., 2023). In terms of that, this paper could be seen as a simple attempt to borrow the idea into another form of NVS, OSRT. While the video diffusion-like multi-frame architecture for considering consistency might be a novelty, the ablation test or comparisons on the decoder architecture are missing.
    - While the core idea is straightforward, some implementation details or hyperparameters for better performance may be informative and implicitly have novelty if the codebase is released. Do you have any plans to release it publicly?
- 3D inconsistency. SRTs do not have 3D consistency in design. Furthermore, DORSal could have worse 3D inconsistency due to its decoding process, in my understanding. From supplementary rendering videos, DORSal's results seem a little unstable when changing viewpoints. An example of objects changing their shapes continuously is shown in the center video of dorsal_multishapenet_3.gif.
    - If possible, more investigation on 3D consistency is nice (while I understand the evaluation protocol is not obvious depending on the settings). The current test (only Sec. 5.3's) is quite limited and might underestimate failure cases by DORSal.
    - (It could be a nice defense to suggest NVS applications requiring less 3D consistency if they exist.)


----

Additional citations
- Generative Novel View Synthesis with 3D-Aware Diffusion Models, Eric R. Chan et al., ICCV 2023
- Diffusion with Forward Models: Solving Stochastic Inverse Problems Without Direct Supervision, Ayush Tewari et al., NeurIPS 2023

**Questions:**

- Conditioning for diffusion models uses frozen OSRT's representations. Is freezing required for better performance? Any experiments on joint training or two-stage training (i.e., 1. training OSRT -> 2. training OSRT and diffusion)? Appendix A says, "End-to-end training comes with additional challenges (e.g. higher memory requirements), but is worth exploring in future work." Is the memory requirement extremely high?
- Can DORSal faithfully reproduce images from the observed (input) viewpoints? Although OSRT was not able to reproduce inputs due to its blurred reconstruction, DORSal could do that. Even if it cannot, the result is very informative for analyzing the behavior. It may indicate that DORSal is still very unfaithful in terms of reference ability in addition to 3D consistency or estimation.
- (a little out of interest) Does DORSal work in out-domain scenes? While I guess that OSRTs may not be good at out-domain scenes, DORSal could possibly be more generalized to them thanks to diffusion-based training. If true, it would be a new and significant strength of the DORSal.
- Sec. 3.2 says, "To obtain instance segments from edits with DORSal, we propose the following procedure..." Actually, is the procedure also used for OSRT baselines in comparison? Or did baselines use different methods?
- Sec. 5.3. says, "This is also reflected in our quantitative ... mixed views." This description seems unclear. Could you provide the details of the test procedure? Is PSNR calculated by a rendered view and a +360 degree re-rendered view?
- Fig. 6 says, "Notably, the encircled tree in the final row is generated upon removal of a slot to fill up the now-unobserved facade previously explained by this slot. The original scene does not contain a tree in this position." Could you explain this more? I didn't understand whether this was success or failure (I felt it was bad behavior).
- Fig. 5 shows combination-based editing. Did the results render only objects which are shown in one of the target scenes? Or, are there many unintended (or unintended-shape) objects? And, does each column is rendered from the same camera pose? In other words, I wonder whether I can assume an object is transferred to the same position (in the image) after the composite. If not, checking the result is pretty difficult.

---

> ### Author Response · Authors · 2023-11-15
> **Response to reviewer dJMA (part 1)**
>
> Thank you for your review and suggestions to improve our work! We are excited that you agree that this work addresses an important problem and is a “nice contribution to the research of SRT”. We were also glad to learn that the paper is easy to read, well-written and very fluent and informative. Thank you also for the additional citations, which we will make sure to include.
>
> In terms of the Weaknesses (W) and Questions (Q) you list, our response is as follows:
>
> **W1: A little reserved technical novelty.**
>
> We mostly agree with your assessment regarding technical novelty, and your summary of our work as a “Straightforward success of the proposal idea to an important problem” is well put. The focus of this work is on making progress towards NVS from few observations in real-world scenes, while also opening up (or preserving, depending on how you look at) the possibility for scene editing. We achieve this by reusing and repurposing existing techniques from the literature, i.e. object-based representations of 3D scenes and video diffusion decoders. In demonstrating the feasibility of this approach (and what’s more, how well it actually fares against the baselines), we expect follow-up work to be able to make further progress toward achieving these ambitious goals.
>
> Because of this, we did not prioritize experiments that specifically target the design of the decoder or other architectural parts. In fact, at the pace at which video diffusion models progress, it is not unreasonable to expect a future version of DORSal to incorporate a slightly different decoder design (e.g. using latent diffusion). In terms of code, there is a publicly available implementation of OSRT (https://github.com/stelzner/osrt) and video diffusion models (https://github.com/lucidrains/video-diffusion-pytorch) already available. Hence, it is unlikely that we will release our own implementation of these. That said, we are happy to update the paper with additional details or hyperparameters if you find that anything is missing. We would also be happy to include pseudo-code in the paper if you think that would be useful. Generally, we have tried to take care in putting as much detail in the paper as possible.
>
> **W2: 3D inconsistency. SRTs do not have 3D consistency in design. Furthermore, DORSal could have worse 3D inconsistency due to its decoding process, in my understanding.**
>
> We acknowledge that DORSal is not perfectly 3D consistent for the same reasons that SRT is not. And indeed, the fact that DORSal is stochastic adds an additional significant challenge, as any new information the model adds during the generation process needs to be synchronized between all generated views. Video diffusion models offer a promising solution for this problem for short, consistent video clips (and in the future likely for much longer videos) by jointly modeling a full sequence of frames. Achieving perfect consistency across a long camera path is still highly challenging as it goes beyond the context length that can be modeled directly via attention in the video diffusion model.
>
> To provide some further insight into the issue of 3D consistency, we have re-computed the edit-segmentation scores on a subset of the samples for DORSal, where we measure both Edit FG-ARI across all the views simultaneously and a “2D Edit FG-ARI” where we compute Edit FG-ARI for each view individually and then average the results. Note that the latter does not penalize inconsistencies between the views, hence the gap between the two scores is indicative of any inconsistencies taking place. A similar approach to evaluating 3D consistency was carried out in the OSRT paper. We obtain 0.702 Edit FG-ARI and 0.721 2D Edit FG-ARI. The small gap between these scores indicates that the segmentations obtained via this procedure are highly consistent across views.
>
> In terms of possible applications, a more 3D consistent approach is typically preferred, and in our paper we have suggested two possible techniques for improving this. Firstly, as shown on MultiShapeNet, having a view distribution that mixes close by and far apart cameras is more desirable. Secondly, having additional stages of “frame shuffling” throughout denoising can help address this. In the limit, where each stage corresponds to a single denoising step, this becomes similar to the “stochastic conditioning” technique deployed in 3DiM for this purpose. Finally, although we haven’t shown this, it is foreseeable that a cascaded diffusion approach using additional super-resolution stages can help resolve tiny inconsistencies.

---

> > ### Author Response · Authors · 2023-11-15
> > **Response to reviewer dJMA (part 2)**
> >
> > **Q1: Conditioning for diffusion models uses frozen OSRT's representations**
> >
> > We refer to our detailed reply to reviewer 16Uh for this, who unfortunately considers this to be a major concern (which we disagree with). To summarize:
> >
> > We have not explored end-to-end training for a variety of reasons:
> >
> > * Joint (end-to-end) representation learning and generative modeling with diffusion models is an active area of research and achieving competitive representation learning performance this way (as opposed to using deterministic reconstruction objectives or contrastive learning) is an open research problem.
> > * We rely on the ability to inject segmentation supervision into the OSRT base model on MSN-H for significantly improved scene editing performance. While this is straightforward for OSRT, where the decoder provides a clear (soft) segmentation prediction, it is unclear how to achieve this when training the OSRT encoder end-to-end through a DORSal diffusion decoder, which does not provide an obvious modeling path for segmentations.
> > * Running such an end-to-end experiment is non-trivial with the existing models used in our work since they live in different code bases and are optimized to make maximum use of available device memory.
> >
> > The memory requirements of DORSal are comparable to that for video diffusion models. Though our implementation leverages data parallelism, we were able to train it on a single device without the need for model parallelism.
> >
> > **Q2: Can DORSal faithfully reproduce images from the observed (input) viewpoints?**
> >
> > That is a great question, and not something we have previously considered evaluating. We ran DORSal and sup-OSRT on 16 scenes of MultiShapeNet and obtained the following results:
> >
> > |      | PSNR  | LPIPS |
> > | ----------- | ----------- | ----------- |
> > | DORSAL      | 19.545141 | 0.2522242 |
> > | Sup-OSRT   | 23.722002  | 0.33968368 |
> >
> > For both models, we observe how predicting input views improves PSNR and LPIPS (compared to their respective novel-view synthesis scores), indicating that the models are better able to leverage the information contained in object slots in this setting. In particular, the significantly lower LPIPS score for DORSal indicates faithful reconstruction of the inputs  This is a positive result for DORSal in particular, suggesting that it hallucinates less in these instances. In the supplementary material (*osrt_dorsal_input_views.pdf*) we now show example renderings. Here it can be seen how Sup-OSRT still produces very blurry predictions, while DORSal generations are sharp. Despite the improvement, and the fact that DORSal generates highly plausible completions, there are also many slight deviations that can be seen. Thank you for suggesting this experiment, we agree it is quite informative and will include it in the paper.
> >
> >  **Q3: (a little out of interest) Does DORSal work in out-domain scenes?**
> >
> > Thank you for making this suggestion. Actually, the test set of MultiShapeNet on which we evaluate is out of distribution already: scenes are guaranteed to contain both novel arrangements as well as novel objects. Generalization to a different domain, e.g. “in-the-wild” web images, would likely require utilizing large-scale pre-trained diffusion models, which is an interesting avenue for future work.
> >
> > **Q4: Sec. 3.2 says, "To obtain instance segments from edits with DORSal, we propose the following procedure..." Actually, is the procedure also used for OSRT baselines in comparison? Or did baselines use different methods?**
> >
> > Yes, we use the same evaluation procedure (pixel-level differences) to evaluate all baselines as well.
> >
> > **Q5: Sec. 5.3. says, "This is also reflected in our quantitative ... mixed views." This description seems unclear. Could you provide the details of the test procedure?**
> >
> > Thank you for pointing this out. In this set-up, we use PSNR as a proxy of 3D consistency for lack of a better approach. We sample 200 views equally spaced around the target scene, and use the first 10 as input views. We sample 190 views for the remaining camera positions and compute the per-frame PSNR. Here DORSal trained with randomly sampled camera poses performs worse than DORSal trained with a mixture of closeby and far away camera poses. The latter forces the model to pay attention to small differences between target views (as opposed to modeling only the larger changes in perspective), which we find qualitatively benefits consistency. Quantitatively we observe an improvement in PSNR as well.

---

> > > ### Author Response · Authors · 2023-11-15
> > > **Response to reviewer dJMA (part 3)**
> > >
> > > **Q6: Fig. 6, Could you explain this more? I didn't understand whether this was success or failure (I felt it was bad behavior).**
> > >
> > > Good question. It is hard to say whether this behavior is good or bad, and we mainly highlighted it as an interesting situation that may arise as a consequence of using a probabilistic approach. Generally, stochasticity is useful when information is lacking, such as when editing a scene by removing object slots. In that sense, the fact that the model fills up the void left by removing some part of the scene with a tree is a good thing. On the other hand, our goal was to remove information (here: part of the building) from the scene, which means that this is unintended, or at least unexpected behavior.
> > >
> > > There are likely two issues at play here: (1) removing slots may put the model in a regime that is too far out of distribution, and (2) the scene is not perfectly object-decomposed by OSRT, which makes the effect of removing a slot less predictable and adds to this. We expect future iterations of these models to address these unexpected behaviors by mitigating distributional shifts when making changes to the scene. We will clarify this in the paper.
> > >
> > > **Q7: Fig. 5 shows combination-based editing. I wonder whether I can assume an object is transferred to the same position (in the image) after the composite. If not, checking the result is pretty difficult.**
> > >
> > > Thank you for pointing this out. Indeed, it is currently difficult to compare the position of the object in the first scene to its combination, because we use a different camera position (i.e. the camera position provided for this scene in the dataset). In the current visualization, only the camera position between the views in Scene 2 and the Combination are aligned. For example, we can see how the car (encircled in orange) stays at the same location after transplanting and combining it with a different background. The same holds true for the planes in the top row (one of them is encircled in blue), which can be observed in the same location in the second and third view in the Combination scene. Though this provides ample evidence that objects stay in the same location, even if they are combined with background from another scene, we agree it would be better to view Scene 1 from the same camera position. We will make this change in the next revision.

---

> > ### Comment · Reviewer_dJMA · 2023-11-16
> > **Thank you**
> >
> > Thank you for your careful replies and additional information! They deepened my understanding.
> >
> > W1:
> > Thank you. One more thing is that it is nice to add the two diffusion x NVS papers in related work to clarify novelty (and what is not novelty).
> > - Generative Novel View Synthesis with 3D-Aware Diffusion Models, Eric R. Chan et al., ICCV 2023
> > - Diffusion with Forward Models: Solving Stochastic Inverse Problems Without Direct Supervision, Ayush Tewari et al., NeurIPS 2023
> >
> > W2:
> > The experiment on 3D consistency is quite helpful. While the setup may underestimate 3D inconsistency because 3D consistency of appearance (maybe, typically referred to as "3D consistency") is more sensitive than that of segmentation, its result is great for the paper.
> >
> > Q1:
> > I understand the issues with the resources and the separate implementations. I hope that future work challenges it.
> >
> > Q2:
> > That is a great experiment! While the input-view scores are better than the novel-view scores (in Table 1), I had expected that even PSNR was much higher. I began to guess that the frozen OSRT representations had already failed to maintain input-view information a lot, so DORSal also cannot reproduce the input views in detail and (almost) deterministically.
> > Anyway, thank you for providing the result!
> >
> > Q3, 4, 5, 6, 7:
> > Thank you!

---

> > > ### Author Response · Authors · 2023-11-22
> > > **Thank you and a question**
> > >
> > > Thank you for your constructive feedback and for engaging during the rebuttal stage. We will include these two papers in the related work section and discuss them. We note that we cite the Chan et al. (2023) paper already, though we are happy to expand on the precise connection. Indeed, although these papers also explore diffusion for NVS, they can be viewed through the lens of NeRFs (as opposed to SRT-based models). Further, they primarily show results for single-object scenes (though Chan et al. (2023) includes results for some indoor scenes) and it remains unclear how to manipulate generated scenes at the object level.
> > >
> > > Unfortunately we have sadly yet to hear back from any the other reviewers and based on the initial scores we are worried that this work will not be given due consideration during the reviewer discussion stage. Since you clearly understood our contribution well (both the strengths and weaknesses), and seem quite positive about our contribution overall, may we ask that you engage with the other reviewers during the discussion stage to share your perspective?

---

> > > > ### Comment · Reviewer_dJMA · 2023-11-23
> > > > **I see**
> > > >
> > > > Fortunately, after the comments, you got new responses from two of the reviewers. Given them, I hope that a well-considered decision will be made by the AC (and us). Thank you!

---

### Official Review · Reviewer_jJcr · 2023-10-31

**Soundness:** 3 good
**Presentation:** 3 good
**Contribution:** 3 good
**Rating:** 6
**Confidence:** 3

**Summary:**

This paper considers an object-centric representation for efficient scene representation and rendering. The work builds on earlier work on object based scene representations (OSRT) but considers a different decoder based on video diffusion models instead of a simple decoder network trained with an l2 loss. Experiments show that the novel view synthesis is much sharper leading to a lower FID score, and that the representation allows for scene editing.

**Strengths:**

- The methodology is sound, by using a video diffusion model for the decoding/rendering which results in sharp and consistent images. This addresses issues with previous works where the renderings in general are quite blurry and/or not consistent.
- Experiments show that the method can obtain significantly lower FID (but not PSNR and LPIPS) for novel view synthesis compared to existing work based on scene representations, although not as significant when compared to existing methods based on diffusion models (3DiM).
- The renderings of the scenes appear consistent across views and over time. Furthermore, for scene editing, the filled in regions when objects are removed appear realistic, and the object slots seem to mostly correspond to specific objects (e.g. a car or a tree) in the scene.

**Weaknesses:**

- The scene editing is more a property of OSRT than the proposed method. The object-slots are pre-trained from OSRT and not refined or learned in this paper.
- One main limitation of the method is that we can not control specific object slots. If we transfer a slot from one scene to the other, we can not (to my understanding, please correct if it is incorrect) e.g. move or rotate it in an easy way. It is placed in exactly the same position as in the original scene, which in general is not very useful.
- The videos in the supplementary material would have been more informative if they would have shown results for all methods compared to as well, and also the input image/images.

**Questions:**

See weaknesses

---

> ### Author Response · Authors · 2023-11-15
> **Reponse to reviewer jJcr**
>
> Thank you for your review and suggestions to improve our work! We appreciate that you recognize that our methodology is sound, that DORSal can obtain significantly better FID, that the renderings appear consistent across views and over time, and that the presented edits are mostly realistic.
>
> **W1: The scene editing is more a property of OSRT than the proposed method. The object-slots are pre-trained from OSRT and not refined or learned in this paper.**
>
> We agree that the editing capabilities we present here are mainly a consequence of having object slots. However, other aspects of the model play an important role too. For example, because DORSal is a probabilistic approach, it is better equipped to handle the inconsistencies that can arise due to removing or combining different slots and the uncertainty that arises from this. Indeed, compare the plane in the middle frame in the top row in Scene 2 to the same frame in the Combination scene directly to the right. Notice how DORSal generates plausible looking shadows under the plane to match the lighting when viewed from this angle as in Scene 1. This impressive feat is a direct consequence of using a diffusion-based decoder and not something that the deterministic OSRT decoder can achieve. Indeed, this observation alone already might indicate that a diffusion-based decoder might be more suitable for the editing operations presented here.
>
>
> **W2: One main limitation of the method is that we can not control specific object slots**
>
> Indeed, we acknowledge that this is a limitation of our current approach and supporting more fine-grained scene editing operations is an important direction for future work. For the rotation and translation you mentioned in particular, it is foreseeable how supervised co-training with language can provide an interface for this as in “Object 3DIT” (Mochel et al., 2023), which we cite in our work. Alternatively, the object representations themselves could be disentangled to a point, where information about the rotation or position of an object is isolated, and can thus be manipulated independently during generation. In general, fine-grained, object-level control is an active area of research, and approaches that offer such capabilities like DisCoScene (Xu et al., 2023) requires a lot more prior knowledge about the scene to be able to work.
>
>
> We will improve our current discussion of limitations to put more emphasis on this, and highlight ways in which future research may address this.
>
> **W3: The videos in the supplementary material would have been more informative if they would have shown results for all methods compared to as well, and also the input image/images.**
>
> Thank you for pointing this out. We will add video results for baselines to the supplementary results in the updated version of our paper. In the meantime, we recommend viewing the website of the OSRT baseline (https://osrt-paper.github.io/) for representative video results.

---

> > ### Comment · Reviewer_jJcr · 2023-11-22
> >
> > Thank you for the answers. I keep my rating as 6, and below are some brief comments.
> >
> > W1: I agree with the raised points that a probabilistic decoder can e.g. handle inconsistencies better, but the possible scene edits are still exactly the same as the baseline method, but with better/more consistent renderings which somewhat limits the technical novelty of the paper.
> >
> > W2: These are all good and valid suggestions, but as pointed out by reviewer J43q it would require significant modifications of the proposed method.
> >
> > W3: Indeed, the videos look less sharp than DORSal. It seems clear from the paper that the proposed method is better qualitatively and by the evaluated metrics.

---

### Official Review · Reviewer_16Uh · 2023-11-04

**Soundness:** 3 good
**Presentation:** 3 good
**Contribution:** 2 fair
**Rating:** 5
**Confidence:** 5

**Summary:**

This paper presents DORSal, a 3D scene generation model that leverages object-centric representations. The approach consists of two stages: first, it pre-trains an Object Scene Representation Transformer (OSRT) to encode multi-view images into slot-based representations that capture the objects in the scene. Second, it trains a diffusion-based conditional multi-view decoder that takes the frozen slot representations as input and renders novel views of the scene. The paper demonstrates that DORSal can generate 3D scenes with higher image quality than the baseline models, and also perform scene manipulation tasks such as object removal.

---

post rebuttal:

rating 3 -> 5

**Strengths:**

1. The generation quality of the proposed model is much better than the baseline models. This is a meaningful improvement as it can help scale up object-centric learning to large-scale applications.
2. The experiments in object-level scene manipulations are quite interesting. The paper demonstrates that the learned representation allows object-level scene editing by removing or transferring slots between scenes. This is a promising result that could lead to further research in this area.

**Weaknesses:**

1. The model reliance on pre-trained object-centric representations instead of end-to-end training with diffusion models is a potential weakness.
    * The quality of the object slots provided by OSRT may not capture the object representation, including the appearance information, well. Eventually, as the decoder becomes stronger and stronger, the quality of the slot representations will become the bottleneck of improving the generation quality. This brings limitation to the model in scaling up to more realistic scenes.

2. Given that the representations are pre-trained, the model seems to be too straightforward and limited in its technical contribution.
    * The improvement of image quality is obvious when the slots are pre-computed and frozen, one might expect the introduction of the diffusion decoder to have some effect on the representation learning process. However, the paper was not able to demonstrate this.

**Questions:**

1. What are the benefits and drawbacks of training DORSal end-to-end? How would it affect the quality of the object representations?
2. In cases where end-to-end training poses challenges, such as collapsing issues, could it achieve better performance by finetuning the pre-trained slots encoder during the decoder training stage?

---

> ### Author Response · Authors · 2023-11-15
> **Response to reviewer 16Uh (part 1)**
>
> Thank you for your review and for your constructive comments. We appreciate that you recognize the strengths of our method, in particular the substantial improvement in generation quality compared to prior work, and the novel conceptual contribution of object-level scene editing in diffusion generative models enabled by conditioning on structured, object-centric scene representations (“object slots”).
>
>
> We address your comments regarding your highlighted weaknesses of the paper in the following.
>
>
> **W1: The model reliance on pre-trained object-centric representations instead of end-to-end training with diffusion models is a potential weakness.**
>
>
> We’d like to respectfully push back on this argument (in the context of this being a weakness of our paper), for the following reasons:
> * Joint (end-to-end) representation learning and generative modeling with diffusion models is an active area of research and achieving competitive representation learning performance this way (as opposed to using deterministic reconstruction objectives or contrastive learning) is an open research problem. Solidly addressing this problem goes far beyond the scope of our investigation. In the future, as the community makes progress on this problem, e.g. in the setting of single-image generative models, it would be great to revisit the end-to-end setting in the context of DORSal.
> * We rely on the ability to inject segmentation supervision into the OSRT base model on MSN-H for significantly improved scene editing performance. While this is straightforward for OSRT, where the decoder provides a clear (soft) segmentation prediction, it is unclear how to achieve this when training the OSRT encoder end-to-end through a DORSal diffusion decoder, which does not provide an obvious modeling path for segmentations.
> * Separating the problem into a representation learning stage for conditioning information (here: object slots) is common practice in text-conditioned diffusion models. It has several benefits: 1) significantly reduced memory, compute, and infrastructure complexity; 2) modularity: our two-stage training recipe has the advantage of directly benefiting from future advances in scene representation learning methods as well as diffusion generative models.
>
> Regarding your comment “the quality of the slot representations will become the bottleneck of improving the generation quality”: this is true irrespective of whether the model is trained end-to-end with a diffusion-based decoder or a deterministic decoder. We believe that there is no reason to assume that a diffusion-based decoder would necessarily result in better object representations compared to a deterministic decoder (which we chose). Running such an end-to-end experiment is non-trivial with the existing models used in our work since they live in different code bases and are optimized to make maximum use of available device memory (in addition to the issue of segmentation supervision injection). We leave this investigation for future work, once joint representation learning and generative modeling with diffusion models has advanced as a research field.
>
> **W2: Given that the representations are pre-trained, the model seems to be too straightforward and limited in its technical contribution.**
>
> We agree that the novelty solely in terms of architecture contribution for the individual method components is limited. In our view, the novelty and significance lies in the elegance/simplicity of the combined approach (which has not been demonstrated before), as our recipe allows for (almost) direct use of established components and thus also benefits directly from future development in these areas. Indeed, we demonstrate that conditioning a video diffusion architecture on (structured) scene representations and pose encodings allows us to overcome limitations of a prior state-of-the-art 3D generative diffusion model (3DiM) and introduce a sampling process to achieve improved consistency for sampling long camera path videos (up to 200 frames), which is an extremely difficult task.
>
> Aside from NVS, consistent object or asset transfer between scenes in video/3D generative models is a major unsolved problem, and DORSal shows promising progress on this problem. DORSal demonstrates that this recipe can achieve new forms of control (object removal, object transfer) via simple manipulation of the conditioning information without the need to collect paired editing data or the need to perform pixel-level masking and inpainting.
>
> We would appreciate it if you could reconsider your assessment regarding novelty (and significance to the community) to focus on aspects beyond architectural novelty, as discussed above.

---

> > ### Author Response · Authors · 2023-11-15
> > **Reponse to reviewer 16Uh (part 2)**
> >
> > **Q1: What are the benefits and drawbacks of training DORSal end-to-end? How would it affect the quality of the object representations?**
> >
> > See our response under W1.
> >
> > **Q2: In cases where end-to-end training poses challenges, such as collapsing issues, could it achieve better performance by finetuning the pre-trained slots encoder during the decoder training stage?**
> >
> > Fine-tuning the encoder during the diffusion modeling stage is an interesting idea and we agree that it would be great to explore this and other aspects of end-to-end joint representation learning and generative modeling in future work. As end-to-end training would require significant changes to DORSal as mentioned above (unclear how to inject segmentation supervision, device memory constraints in particular for the video diffusion model, infrastructural challenges), we believe that this is beyond the scope of this submission.

---

> ### Comment · Reviewer_16Uh · 2023-11-23
> **Thank you for the response**
>
> I thank the authors for their response and apologies for the delay.
>
> To clarify, my question did not pertain to the comparison between representation learned by diffusion model or OSRT, which would indeed be beyond the scope of this paper. Rather, I am concerned whether the information contained in the pre-trained, frozen representations was sufficient for the diffusion decoder to accurately render the fine details in image generation. This is questionable given the fact that the representation learning approach, OSRT, demonstrates suboptimal reconstruction quality in complex images. This could suggest that it might not capture the finer details necessary for generating new viewpoints of the same objects later when using the diffusion model.
>
> However, after reviewing the feedback from other reviewers and the authors' responses, I do agree that it is interesting to see that the high-quality image generation can be achieved by the simple combination of existing approaches. Moreover, it is also quite surprising to see that the frozen representations could provide generations that are far beyond the detail level demonstrated by the results of the representation learning model OSRT.
>
> Considering these points, I am raising my rating.

---

### Author Response · Authors · 2023-11-22
**General comment**

We thank the reviewers for their comments and constructive feedback, which has helped us improve the paper. Generally, there appears to be broad agreement regarding the effectiveness of our approach and the presentation quality of the paper. To highlight a few strengths listed by the reviewers, which we agree summarize the contribution well:

* “​​The generation quality of the proposed model is much better than the baseline models. This is a meaningful improvement [...]” (`16Uh`)
* “The experiments in object-level scene manipulations are quite interesting” (`16Uh`)
* “The methodology is sound [...] which results in sharp and consistent images. This addresses issues with previous works” (`jJcr`)
* “The renderings of the scenes appear consistent across views and over time” (`jJcr`)
* “Straightforward success of the proposal idea to an important problem” (`dJMA`)
* “This is a nice contribution to the research of SRT because the SRT family has been likely to render very blurred images, which is one of the largest weaknesses.” (`dJMA`)
* “The paper demonstrates the effectiveness of using an object-centric representation for controllable scene synthesis” (`J43q`)
* “Training a multi-view diffusion model conditioned on such representation can [...] enable many applications like object removal and transferring.” (`J43q`)

Based on reviewer feedback, we have conducted additional experiments to assess the 3D consistency of DORSal (`dJMA`), the degree to which it hallucinates on input views (`dJMA`), and provided several other requested materials such as OSRT decompositions on Street View (`J43q`). Meanwhile, we are in the process of revising the paper to include your comments about related work, and a better discussion of limitations, such as fine-grained object-level scene control. The changes to the text are generally expected to appear at a later stage, though please let us know if you would like to a see the fully revised draft ahead of time.

---

### Meta-Review · Area_Chair_oy8N · 2023-12-07

**Metareview:**

The paper proposes an approach for 3D (multi-view, to be precise) scene generation which conditions on object-centric representation and thus support object-level editing to some extent. The paper receives three positive and one negative ratings. The positive reviews are based on the effectiveness of the proposed framework. Also, the statements are well-corroborated and the visual results are good. The negative reviews concerned the technical novelty as the framework heavily rely on the quality OSRT and diffusion models. However, quoting comments from Reviewer dJMA, it is a "Straightforward success of the proposal idea to an important problem". I believe it is a valuable work to the community.

**Justification For Why Not Higher Score:**

The work makes a steady progress toward the 3D view synthesis problem. Despite solid and effective, the proposed framework builds upon mostly existing components, which to some extent limit the scale of contributions.

**Justification For Why Not Lower Score:**

The proposed method is sound and well corroborated.

---

### Decision · Program_Chairs · 2024-01-16

Accept (poster)